

# Transformation of logwood combustion emissions in a smog chamber: formation of secondary organic aerosol and changes in the primary organic aerosol upon daytime and nighttime aging

Tiitta, P.[1], Leskinen, A.[2,3], Hao, L.[2], Yli-Pirilä, P.[1,2], Kortelainen, M.[1], Grigonyte, J.[1], Tissari, J.[1], Lamberg, H.[1], Hartikainen, A.[1], Kuuspalo, K.[1], Kortelainen, A.[2], Virtanen, A.[2], Lehtinen, K. E. J.[2,3], Komppula, M.[3], Pieber, S.[4], Prévôt, A. S. H.[4], Onasch, T. B.[5], Worsnop, D. R.[5], Czech, H.[6], Zimmermann, R.[6,7,8], Jokiniemi, J.[1], and Sippula, O.[1,8]

[1]Department of Environmental and Biological Sciences, Univ. of Eastern Finland, P.O. Box 1627, 70211 Kuopio, Finland
[2]Department of Applied Physics, Univ. of Eastern Finland, P.O. Box 1627, 70211 Kuopio, Finland
[3]Finnish Meteorological Institute, P.O. Box 1627, 70211 Kuopio, Finland}
[4]Laboratory of Atmospheric Chemistry, Paul Scherrer Institute, Villigen, Switzerland
[5]Aerodyne Research, Inc., Billerica, MA 08121, USA
[6]Joint Mass Spectrometry Centre, Univ. at Rostock, Institut für Chemie, Lehrstuhl für Analytische Chemie, Dr.- Lorenz-Weg 1, 18059 Rostock, Germany
[7]Joint Mass Spectrometry Centre, Cooperation Group Comprehensive Molecular Analytics, Helmholtz Zentrum München, Germany
[8]HICE-Helmholtz Virtual Institute of Complex Molecular Systems in Environmental Health−Aerosols and Health (www.hice-vi.eu)

*Correspondence to*: P. Tiitta (petri.tiitta@uef.fi)

**Abstract.** Secondary organic aerosol (SOA) derived from small-scale wood combustion emissions are not well represented by current emissions inventories and models, although they contribute substantially to the global atmospheric particulate matter (PM) levels. This type of SOA is formed via the oxidation of combustion-emitted volatile organic compounds, inducing large changes in the properties of combustion-derived PM. In this work, a 29 m$^3$ smog chamber in the ILMARI facility of the University of Eastern Finland was deployed to investigate the formation of SOA from small-scale wood combustion. Emissions were oxidatively aged in the smog chamber for a variety of dark (i.e., O$_3$ and NO$_3$) and UV (i.e., OH) conditions, with OH concentration levels of (0.5–5)×10$^6$ molecules cm$^{-3}$, achieving equivalent atmospheric aging of up to 18 hours. A soot particle aerosol mass spectrometer (SP-ToF-AMS) was applied to measure real-time concentrations of the major chemical mass fractions of submicron particles in chamber conditions simulating ambient wood combustion plumes. Substantial SOA formation was observed in every experiment performed with three wood species (birch, beech and spruce) and two ignition process (fast ignition with a VOC-to-NOx ratio of 3 and slow ignition with a ratio of 5). Both dark and UV aging increased the SOA mass, and OA enhancement resulted in an average SOA production of 2.0 times the initial OA mass loadings. The enhancement was found to be higher for the slow ignition compared with fast ignition. UV aging revealed faster SOA formation rates than dark aging.





OA elemental analysis indicated that the oxidatively aged wood combustion emissions exhibited Van Krevelen slopes of approximately -0.7, with slightly steeper slopes for the fast ignition experiments than for the slow ignition cases. These results are, in Van Krevelen space, in the same region as measured ambient OA, and the OA aging was found to follow a similar slope with ambient observations, suggesting that our chamber experiments are well suited to simulate polluted boundary-layer

conditions. To investigate SOA composition in detail, positive matrix factorization (PMF) was used for separating SOA, POA, and their subgroups from the total OA spectra. This resulted in two POA factors (POA1 and POA2) and three SOA factors (SOA1, SOA2, and SOA3) representing the three major oxidizers: ozone, the nitrate radical and the OH radical. The SOA of the first factor (SOA1) was likely formed via ozonolysis of unsaturated compounds, whereas SOA2 contained secondary organonitrates from nitrate radical oxidation. The third SOA factor (SOA3) was indicative of SOA formation dominated by

the OH radical.

The PMF results showed that most POA was oxidized after the ozone addition, forming aged POA, and after 7 h of aging (dark + UV aging), more than 75% of the original POA was transformed. This process may involve evaporation and homogeneous gas-phase oxidation as well as heterogeneous oxidation of particulate organic matter. The results generally prove that logwood burning emissions are subject of intensive chemical processing in the atmosphere, and the time scale for these transformations

is relatively short, i.e., hours.

## 1 Introduction

Small-scale wood combustion emits large amounts of submicron black carbon (BC), organic aerosol (OA) (e.g., Lanz et al., 2010; Crippa et al., 2014), PAHs (e.g., Orasche et al., 2012; Eriksson et al., 2014; Bruns et al., 2015), and gaseous pollutants, such as $NO_x$ and volatile organic compounds (VOCs) (e.g., McDonald et al., 2000) into the atmospheric boundary layer. These

emissions cause adverse health effects, reduced visibility and affect the climate. The main properties of soot particles, which are mixtures of elemental carbon and internally mixed organic and inorganic species (ash), may change significantly due to coatings formed by atmospheric aging of emissions. The aging processes have been shown to affect particle hygroscopicity and scattering (e.g., Cappa et al., 2012) and their toxicological properties (e.g., genotoxicity, Nordin et al., 2015). Thus, atmospheric aging processes are likely to significantly affect the key properties of combustion emissions with respect to health

and climate. A better understanding of the formation processes of aerosol during aging in polluted air and combustion plumes is necessary for assessing atmospheric influence of combustion aerosol and their health effects.

The organic species emitted from wood combustion are either in the gas phase or condensed phase which is mainly present as a layer of adsorbed/condensed organic matter on primary particles. These primary particles are mainly BC agglomerates in small-scale batch-wise operated appliances (Torvela et al., 2014; Leskinen et al., 2014). The condensed coatings on BC

particles consist of a complex mixture of compounds formed via the thermal degradation of wood polymers and their further reaction products. These include monosaccharide anhydrides, methoxyphenols, polyaromatic hydrocarbons (PAHs),





functionalized PAHs, and carboxylic acids (Hayes et al., 2011; Orasche et al., 2013) with different volatilities (i.e., semi-volatile, intermediate volatile, and low-volatile particles). Semi- and intermediate volatile compounds favor partitioning into the gas phase when diluted or exposed to higher temperatures (e.g., Donahue et al., 2006; Kroll and Seinfeld 2008). Due to the presence of semi- and low-volatile organics in the emissions, the freshly emitted particles from wood combustion are coated

with organic matter (Tissari et al., 2008, Torvela et al., 2014), although the coating is likely to substantially increase due to the condensation of secondary organic matter formed either from anthropogenic or from biogenic organic precursor gases in the atmosphere (e.g., Akagi et al., 2012).

VOCs can undergo chemical transformation by three major oxidizers: ozone, the OH radical, and the nitrate radical (e.g., Kroll and Seinfeld 2008; Chacon-Madrid and Donahue 2011). When OH concentrations are low, ozone and/or nitrate radicals

dominate the homogenous oxidation of VOCs to form less volatile organic compounds; for example, double bonds of alkenes may be broken by ozonolysis to produce functionalized lower-vapor-pressure products compared to the VOC precursors or fragment to smaller gas-phase VOCs. When sufficient UV radiation is available, OH radicals are produced by photolysis and dominate the oxidation of VOCs because of their high reactivity. These atmospheric oxidation reactions generateproducts with lower volatility which condense on the existing soot agglomerates forming secondary organic aerosol (SOA). If the

condensation sink is sufficiently small after dilution in clean air, new particle formation can also occur.

The emission regulations in the EU (EU commission regulation 2015/1185) focus exclusively on the primary aerosol emissions and neglect the formation of secondary organic aerosol (SOA), which may result in ineffective legislation and actions when aiming to reduce small-scale combustion emissions. There have been many studies about primary emissions from different combustion appliances and fuel types, yielding relatively reliable estimates on the primary emission factors from small-scale

wood combustion (e.g., Sippula et al., 2007; Lamberg et al., 2011; Tissari et al., 2008). Moreover, recent on-line aerosol chemical composition measurements have improved the time resolution (Schneider et al., 2006; Weimer et al, 2008; Heringa et al., 2011; Heringa et al., 2012; Elsasser et al., 2013; Leskinen et al., 2014; Kortelainen et al., 2015) and have emphasized the importance of combustion appliance type and wood quality as well as operational practices of combustion appliances on PM emission quantity and quality.

Field studies have shown that biomass burning aerosol particles undergo significant changes in their properties that are mainly driven by atmospheric oxidation (Vakkari et al., 2014, Reid et al., 2005; Capes et al., et al., 2008; Akagi et al., 2012). However, results from field studies are not easily comparable because of variations in uncontrolled emissions and conditions. Thus, detailed investigation of SOA formation from biomass burning processes requires well-controlled laboratory studies (e.g., Grieshop et al., 2009a). Regarding open biomass burning, the study of Ortega et al. (2013) reported SOA formation in a flow

reactor from biomass burning smoke samples which were generated in an open fire chamber with 25 different types of biomass. An average SOA formation of 1.4 times the initial POA was observed, with SOA concentrations peaking after three days of photochemical age ($OH_{exp} \sim 3.9 \times 10^{11}$ molec. cm$^{-3}$s). Similarly, the study of Hennigan et al. (2011) reported an average OA



mass enhancement ratio of 1.7 (varying from 0.7 to 2.9) in smog chamber experiments with biomass typical in North American wildfires.

Regarding residential wood combustion processes, several laboratory studies have been conducted to determine SOA emission factors and chemical composition (e.g., Corbin et al. 2015a) using chambers or continuously operated flow reactors. Photo-oxidation experiments with wood batch combustion emissions in smog chambers have revealed that wood-burning net SOA-to-POA ratios are at least 1.5 or higher (Grieshop et al., 2009a; Heringa et al., 2011, Bruns et al., 2015). In a previous study on a modern masonry heater (Bruns et al. 2015), the emissions were aged in a 29 m$^3$ smog chamber for 4.5–16 h, corresponding to OH exposures of $(0.7–2) \times 10^{11}$ molec. cm$^{-3}$s. They reported an OA increase by factors of 3 and 1.6 for high and low stove loadings of beech logs, respectively. They also found that the contribution of the total average PAH to the total OA was 4 % and 15 % for low and high loadings, respectively, and that functionalized PAHs increased with ongoing aging.

Small-scale combustion emissions have received insufficient attention until today, although this type of emission is known to be among the largest sources of OM (organic mass) and climate-active BC (e.g., Denier van der Gon et al., 2015). These emissions typically increase in cold weather because of the increased use of wood combustion, which leads, together with a suppressed atmospheric mixing layer (e.g., inversion) and relatively low emission height, to peaks in ambient boundary-layer aerosol concentrations in regions where wood burning is common (Saarikoski et al., 2008; Yli-Tuomi et al., 2015). To investigate the transformation of these emissions, the effects of both dark and photochemical aging on the emissions from wood combustion were studied in a smog chamber (Leskinen A. et al., 2015) in this work, simulating the dilution and chemical reactions occurring in a combustion plume under boundary-layer conditions. The used combustion appliance was a typical, modern Northern European masonry heater (Nuutinen et al., 2014) fired with spruce logs. The specific aims were to 1) quantify the emission factors of SOA from a representative logwood-fired masonry heater, 2) characterize the evolution of OA chemical composition during aging, and 3) investigate the effects of different oxidation mechanisms on SOA formation.

## 2 Methods

### 2.1 Procedure of emission aging experiments

Atmospheric aging processes of log wood combustion emissions were studied in the ILMARI facility of the University of Eastern Finland (http://www.uef.fi/en/web/ilmari/home). The experiments focused on investigating the effects of dark and UV-light induced aging of residential-scale wood combustion emissions and were carried out in a 29 m$^3$ smog chamber (Leskinen A. et al., 2015). The experimental conditions are listed in Table 1, and the setup is presented in Figure 1.

A modern heat-storing masonry heater with a staged combustion air supply (Leskinen J. et al., 2014; Reda et al., 2015) was used as the combustion source. In each experiment, 2.5 kg of wood logs (spruce [*Picea abies*], birch [*Betula pubescens*] and beech [*Fagus sylvatica*]) were burned with combustion initiated from 'cold start', i.e., the firebox was at room temperature. Ten logs (~235 g each) were laid crosswise with 150 g of smaller wood pieces as kindling on top. The combustion was initiated





by igniting a batch of wood sticks, serving as kindlings, on top of the wood batch. Low pressure in the stack was adjusted with a flue gas fan and dampers to $12.0 \pm 0.5$ Pa below ambient pressure. The moisture content of the wood logs was 7.4 % (See Reda et al. 2015 for details). The ignition of the wood batch was carried out in two different ways: 1) smaller kindling size resulted in faster ignition of the main batch and 2) larger kindling size (double) resulted in slower ignition of the main batch

(Table 1). The mass of kindling was 150 g / ignition. The gaseous emissions from the raw exhaust gas were measured online with a multicomponent FTIR analyzer (Gasmet), including $CO_2$, CO, NO, $NO_2$, $CH_4$ and 15 typical hydrocarbon compounds (Table S1, Figs. S1 and S2) for wood combustion emissions. In addition, the analyzer was equipped with an $O_2$ sensor.

The combustion exhaust was sampled with a system in which the sample is drawn though a PM10 pre-cyclone, a heated probe (150 °C), and then diluted in two stages. The primary dilution was conducted with a porous tube diluter, and the secondary

dilution with an ejector diluter (DI-1000, Dekati Ltd.). Clean air at ambient temperature was used as the dilution gas, and the sampling line dilution ratio was adjusted to 24–31 for samples entering the chamber. The sampling was performed in a pre-filled chamber with clean air such that the total chamber dilution ratios ranged from 229 to 456 (Table 1). The sampling lines from the porous tube diluter to the environmental chamber were heated to 100 °C and externally insulated to reduce line losses. The dilution ratios in the sampling line and in the chamber were monitored by following the $CO_2$ concentrations of the raw

combustion gas, pre-diluted sample (ABB $CO_2$ analyzer), and in the chamber (Vaisala GMP 343, Finland). The smog chamber is a collapsible Teflon bag chamber that is equipped with 47 blacklight lamps (350 nm, Sylvania F40 W/350 BL) (Leskinen et al., 2015). Before each experiment, the chamber was emptied for cleaning, filled with purified air (Model 737-250, Aadco Instruments Inc., USA) with a relative humidity set to approximately 50% (Model FC125-240-5MP, Perma Pure LLC., USA), and finally flushed overnight. The average temperature and relative humidity in the chamber during each experiment was 18

$\pm$ 2 °C and $60 \pm 5\%$, respectively.

In the first set of experiments, the POA and dark aging products of different wood species were compared (Exp. 1A-3A). These experiments were carried out with spruce, birch, and beech logs. In the second set of experiments, a detailed investigation was executed with spruce logs as fuel (Exp. 1B-5B); these experiments included aging both via dark aging and via UV-light induced photochemical reactions. In experiments 1B-3B, dark aging was followed by OH exposure (photo-oxidation), whereas

simultaneous $O_3$ and OH exposure was conducted in experiments 4B and 5B. The experiments began with the filling of the chamber with a prediluted sample of the first batch of log wood combustion emissions (35 minutes). During a stabilization period (40 minutes) followed by filling, the sample was mixed; the organics presumably reached an equilibrium based on the $CO_2$ levels and wall-loss-corrected OA time trends. After the stabilization, 1 µL of butanol-d9 (9 ppb) was injected into the chamber to determine the sample OH exposure and photochemical age during the experiments (Barmet al., 2012). The butanol-

d9 decay was monitored by a high-resolution proton transfer reaction mass spectrometer (PTR-TOF 8000, Ionicon Analytik, Innsbruck, Austria). During each experiment, all of the NO in the chamber was converted to $NO_2$ by adding $O_3$. After the full conversion, an additional 40 ppb of $O_3$ was added. After 3 h of dark aging, the UV lights were switched on and the aerosol particles were aged with UV for 3–4 h (Exp. 1B-3B), corresponding to total OH exposures of $(4–6) \times 10^{10}$ molec. $cm^{-3}s$. In



addition, in experiment 3B, HONO was injected into the chamber after the dark aging to simulate HONO-rich conditions in the atmosphere. The hydroxyl (OH) radical is one of the main reactive species in the atmospheric boundary layer; peak daytime OH radical concentrations are in the range of $(1\text{-}10) \times 10^6$ molec. cm$^{-3}$ (e.g., Atkinson et al., 2000). Moreover, OH concentrations for typical wintertime China conditions exceed $0.4 \times 10^6$ (Huang et al. 2014). HONO was produced by the

titration of sodium nitrite (NaNO$_2$) solution (1 wt %) into sulfuric acid (H$_2$SO$_4$) solution (10 wt %) in a glass flask equipped with a magnetic stirrer based on the method developed by Taira and Kanda (1990). The gaseous HONO was guided by a purified air flow of 3.0 L min$^{-1}$ into the chamber. In experiments 4B and 5B, UV lights were switched on immediately after ozone injection, corresponding to OH exposures of $(5\text{--}7) \times 10^{10}$ molec. cm$^{-3}$s. A similar procedure was applied for all sets of experiments (1A-3A and 1B-5B) except the first experiments (1A-3A), which were conducted with a shorter stabilization

period (10 minutes) before ozone addition; no injection of butanol-d9 was performed in these experiments.

## 2.2 Measurements of particles and their chemical composition in the chamber

Particle number concentrations and size distributions were measured with a scanning mobility particle sizer (SMPS, CPC 3022, TSI). PM$_1$ filter samples were collected on 47 mm quartz fiber filters (Pallflex, Tissuquartz) from the chamber in the second round of experiments after the injection of the primary emissions and at the end of each period (2−3 filter samples / 

experiment). A pre-impactor (Dekati Ltd.) with a cut-off diameter of 1 μm was applied. The organic and elemental carbon were determined from quartz filter samples by a thermal optical method using a carbon analyzer from Sunset Laboratories according to the National Institute for Occupational Safety and Health (NIOSH) procedure, which is described in Sippula et al. 2007 and 2009.

The soot particle aerosol mass spectrometer (SP-HR-ToF-AMS; Onasch et al., 2012) was applied to measure the major fraction

of submicron particles (Aerodyne Research Inc., USA). The SP-AMS is a high-resolution aerosol mass spectrometer (HR-AMS) (DeCarlo et al., 2006) that is equipped with a laser vaporizer based on single-particle soot photometer technology (SP2, Droplet Measurement Technologies, CO, USA, Stephens et al., 2003). The combination of the thermal vaporizer (600 °C) and the continuous wave (CW) laser vaporizer (1064 nm) enables the study of both non-refractory (NR-PM) and refractory light-absorbing submicron aerosol particles (R-PM, e.g., rBC). The aerodynamic lens with 100% transmission range of 75–650 nm

(Liu et al., 2007) focuses particles into a narrow beam that crosses the laser beam vaporizing particles containing rBC (~4000 K) and subsequently onto a heated conical porous tungsten surface (600 °C) at the center of the ionization chamber. Vaporized molecules are ionized by electron impact (70 eV) and detected with a high-resolution time-of-flight mass spectrometry (Tofwerk AG) in positive ion mode. The HR-TOF-AMS was operated in V-mode (Δm/m ~ 2000) from 12 m/z to 555 m/z, which enables the identification, separation, and quantification of typical organic and inorganic species. Two vaporizer

configurations, i.e., dual laser and tungsten vaporizers and tungsten vaporizer only modes, were alternated every 60 s; particle time-of-flight (PToF) modes were operated for 20 s per minute.



Standard mass-based calibrations were performed for the ionization efficiency (IE) in the AMS using dried and size-selected (300 nm) ammonium nitrate ($NH_4NO_3$) particles (Jayne et al., 2000). $NH_4NO_3$ calibrations allow for the determination of non-refractory species using the standard species-specific relative ionization efficiencies (RIEs). Regal Black (Regal 400R Pigment Black, Cabot Corp.) particles were analyzed to determine the relative ionization efficiency for rBC ($RIE_{rBC}$) with values of

0.16 and 0.25 for the first and second set of experiments, respectively. A detailed description of the SP-AMS is presented in Onasch et al., (2012). The aerosol mass concentrations were also corrected for the particle collection efficiency (CE). The CE values can be smaller than 1 because of losses in the aerodynamic inlet or on the vaporizer (tungsten) or increasing divergence in the particle beam (laser vaporizer). Collection efficiencies (CEs) of 0.5 and 0.6 for tungsten (Canagaratna et al., 2007) and the laser vaporizer (Willis et al., 2014), respectively, were applied in the concentration calculations and verified by comparisons

with Tapered Element Oscillating Microbalance (TEOM, Thermo Scientific, model 1405) total $PM_1$ mass. Any uncertainties associated with the CEs may affect the absolute concentrations of the measured species and the relative OM-to-rBC ratios, although they are less likely to affect the SOA formation ratios, which are discussed later.

### 2.3 Data analysis

Particle wall losses (WL correction) have been characterized for decades (e.g., McMurry and Rader, 1985), although the

estimation of losses of semi-volatile species remains challenging (e.g., Zhang et al., 2014; Kokkola et al., 2014). An incorrect estimation of the WL can cause major uncertainties in SOA yields; therefore, it must be performed carefully. The extent of WL of particles depends on several factors, e.g., the surface area-to-volume ratio, the material of the chamber, electric field and turbulence in the chamber, and particle size and charge distribution. The accuracy of the WL calculations suffers from potential changes in the SP-AMS rBC CE (Willis et al., 2014), unknown wall losses of semi-volatile species (Zhang et al.,

2014), and possible changes in the effective density of soot agglomerates (Leskinen et al., 2014).

In this work, WL corrections were calculated based on the decay in refractory black carbon (rBC) concentrations measured by the SP-AMS and adjusted by the decay of elemental carbon concentrations measured by a thermal-optical method (maximum correction of 15%). The correction based on elemental carbon concentrations was made because of the increased SP-AMS sensitivity for highly coated particles (Willis et al., 2014). The calculated $WL_{rBC,fix}$ values agreed well with $WL_{SMPS}$, which

was calculated from SMPS data (method is described in Weitkamp et al. (2007)) in experiments 4B and 5B, whereas in 1A-3A, $WL_{SMPS}$ exhibited 12–23% higher wall losses. Higher WL corrections would lead to higher SOA-to-POA ratios. WL corrections were applied after the emissions were injected and well mixed in the chamber until the end of the experiments. The PM emissions of the analyzed chemical components were determined by multiplying the measured chamber concentrations with the chamber dilution ratio (DR, Table 1) and the specific emission conversion factor (ECF, $m^3$/kg fuel) determined

according to the standard SFS 5624 (Reda et al., 2015).

AMS data analysis was performed with the analysis software SQUIRREL v1.55C and the standard HR-ToF-AMS data analysis software Peak Integration by Key Analysis (PIKA v1.16 C) adapted in Igor Pro 6.34 A (Wavemetrics). The AMS samples



particles $10^7$ times more efficiently than gases (Canagaratna et al., 2007). Nonetheless, gas-phase $CO_2$ concentrations in the chamber of approximately 500 ppm are sufficiently high to affect the AMS measurements. Thus, the influence of $CO_2$ was corrected using time-dependent gas-phase $CO_2$ measurements and checking corrected data using on-line HEPA filter samples. High-resolution (HR) mass concentrations were calculated using the HR-AMS fragmentation table (Allan et al., 2003; Jimenez

et al., 2003), and j15NN isotope analysis was performed by constraining j15NN with $N_2$. The SP-AMS flow was corrected to standard temperature and pressure, which is consistent with common practice, and the data were post-corrected by the measured air beam (AB). Elemental analysis (EA) (Aiken et al., 2007) was performed by the new method described in Canagaratna et al., (2015).

Positive matrix factorization (PMF) is a statistical tool to determine several characteristic mass spectra (i.e., factors) that best

describe the overall time-dependent mass spectra by minimizing model residuals. User input is required such that the mass spectra factors still have realistic physical meaning (Paatero and Tapper, 1994; Paatero, 1997). Before the analysis, signals were exported using a peak-integration method and the new error matrix model described in Corbin et al., 2015c. The PMF Evaluation Tool v.2.08 was applied, and the standard data pre-treatment process was executed based on Ulbrich et al. (2009), including applying minimum error criteria, down-weighting weak variables based on the signal-to-noise ratio, and m/z 44, NO

and $NO_2$ and water-related peaks. The PMF analysis was performed on the combined high-resolution organic and inorganic nitrate mass spectra (Hao et al., 2014). Particulate nitrate was added to the PMF analysis because the sources of particulate organic nitrate (ON) are not well known (Fry et al., 2009, 2013; Liu et al., 2012; Rollins et al., 2012; Farmer et al., 2010). Moreover, wood combustion is an important source of gaseous nitrogen oxides that can form ON via nitrate radical chemistry (Rollins et al., 2012). Rotational freedom of the solution was investigated with fpeak (a parameter that is used to explorer

ambiguity associated with rotations within the solution space) values between -1 to 1 to minimize the $Q/Q_{exp}$ ratio combined with residuals (unsolved fraction), which was less than 1% in all experiments. Collection efficiency (CE) and relative ionization efficiency (RIE) of 1 is applied (standard procedure) in PMF calculations.

## 3 Results

### 3.1 Primary emissions

The mass spectra of the refractory black carbon (rBC, $C_x^+$) remained constant for all of the aging experiments, with dominating signals at low carbon cluster numbers ($C_1^+$ to $C_5^+$). The $C_1^+$:$C_3^+$ ratio was approximately 0.6, which is similar to the values obtained from a propane diffusion flame (CAST "black") produced by a combustion aerosol standard burner (Corbin et al., 2014). Thus, the logwood materials used in this work (spruce, birch, and beech) suggest a similar elemental carbon chemical structure as obtained by Corbin (2014) and that the microstructural properties do not change remarkably during atmospheric

aging.



In contrast to the elemental carbon, the organic aerosol (OA) on soot agglomerates undergoes intense changes upon oxidation. OA concentrations were well correlated with the SMPS volume (R = 0.8) over the range of oxidation conditions, with a better correlation during dark aging than during UV aging. OA was observed in the same particle size range as rBC, which confirms an increase in organic coatings on soot particles upon aging (internally mixed). The mass ratio of the OA fraction to the refractory black carbon (rBC) ranged from 0.2 (fresh emissions) to 0.4 (aged emissions) (Table 2). The relatively efficient burning conditions in modern masonry heaters and the applied ignition method, in which the ignition is performed on the top of the wood batch with wood sticks, decreased the OA fraction in this study compared to studies conducted using conventional batch combustion appliances (Tissari et al., 2009). Furthermore, well-dried wood fuel was used, which is not always the case in reality. However, the experiments represent an upper limit for the selected combustion appliance and fuel combination because the first batch of logwood combustion typically generates higher emissions than subsequent batches (Leskinen et al., 2014).

The total PM emissions ranged from 0.6 to 1.4 g kg$^{-1}$ (Table 2) for equal wood loading, which agree with previously reported findings (e.g., Tissari et al., 2007, Lamberg et al., 2011; Bruns et al., 2015). Birch showed the highest total emissions and largest rBC mass fractions. Considering only spruce emissions, the fast ignition experiments (Exp. 1B and 4B) with a lower VOC-to-NO$_x$ ratio yielded smaller total emissions than the slow ignition cases (Exp. 2B and 5B) when the VOC-to-NO$_x$ ratios were higher. The potassium (K) concentration, which is one of the main ash components of logwood combustion (Tissari et al., 2007), was measured using the SP-AMS dual vaporizer mode; clear K and j41K signals were observed. However, the potassium emissions are not reported here because potassium analysis using an AMS is subject to additional uncertainties because of the non-standard ionization of potassium leading to significant variation between instruments (Drewnick et al., 2006; Slowik et al., 2010). Using RIE$_K$ of 10 (Slowik et al., 2010), the K emissions were 4.1%, 0.9 %, and 0.4 % of the PM$_1$ emissions for spruce, beech, and birch, respectively. Potassium emitted from wood combustion appears mainly in the form of K$_2$SO$_4$, KCl and K$_2$CO$_3$ (Sippula et al., 2007), which are expected to be externally mixed with soot agglomerates (Torvela et al., 2014). The SMPS data showed that the median diameter of soot agglomerates increased substantially from 70–120 nm to 150–220 nm during aging (Figs. S3 and S4). The number concentrations were in the range of 2.8–6.5 × 10$^4$ # cm$^{-3}$, and the mass closure of the AMS species measurements agreed relatively well with the total PM mass measured with the TEOM (Table 2).

### 3.2 SOA formation

Figure 2 illustrates the formation of SOA during aging in a smog chamber together with particulate NO$_3$ and SO$_4$, ozone, and NO$_x$ concentrations. The results are summarized in Table 3. SOA was calculated using WL-corrected OA concentrations by subtracting the POA mass (average OA at the end of stabilizing phase A). The wall loss correction method is described in Section 2.3. The SOA mass concentration during dark aging was 1.6 and 1.8 times that of the initial POA for beech and birch (Exp. 1A and 2A), respectively, whereas the average ratio was 1.9 (1.6–2.6) for spruce (Table 3) (see also Supplement S2).



Direct measurements of the hydroxyl free radical (OH) concentrations are challenging; therefore, the OH concentration, the related OH exposure, and the so-called atmospheric age in the smog chamber were ascertained by an indirect method based on the decay of butanol-d9 (Barmet et al., 2012). A comparison between lab studies and atmospheric measurements is easier when the age is estimated. In our experiments, the conditions in the chamber were indicative of polluted atmospheric boundary-

layer conditions with an OH concentration of $(0.5–5)\times10^6$ molec. cm$^{-3}$, ozone concentrations of 20–90 ppb and NOx concentrations of 40–120 ppb (Fig. 2, Table 4) (e.g., Atkinson et al., 2000; Lin et al., 2011). HONO was added in experiment 3B, leading to high oxidant and NO$_x$ levels (Fig. 2b).

In experiments 1B and 2B (Fig. 2a), three distinct periods can be identified. First (Fig. 2a: A), logwood emissions were added to the chamber from the first batch (35 min) and subsequently stabilized as identified by constant CO$_2$ and OA concentrations.

Second (Fig. 2a: B), ozone feeding after the stabilization initiated ozonolysis and NO$_3$ radical reactions (e.g., Rollins et al., 2012), leading to the oxidation of VOCs and the formation of low-vapor-pressure reaction products that condensed onto existing soot agglomerates. This generated 1.6 and 1.8 times more SOA mass than the initial POA mass for the fast and slow ignition cases, respectively (Table 3). In the third period (Fig. 2a: C), starting when the UV lights were switched on, rapid growth in SOA was observed. It can be assumed that in this case, the aging reactions were mainly driven by OH radicals,

whose aging times corresponded to 0.6–0.7 days (Table 4) of equivalent atmospheric aging at typical boundary-layer OH concentrations of $1 \times 10^6$ molec. cm$^{-3}$. As a result, the total SOA mass increased to 1.9 (Exp. 1B) and 2.4 (Exp. 2B) times the initial POA mass.

Experiment 3B was similar to 1B and 2B except that HONO (Fig. 2b: H) was added to the chamber. In addition, propene (~5.1 ppm) was injected into the chamber after the HONO injection to adjust the VOC-to-NO$_x$ ratio to a value similar to those of the

original emissions (the ratio after propene injection was 5.7). The products from propene oxidation do not contribute to SOA formation alone (Platt et al., 2013; Leskinen et al., 2014). The SO$_4$ mass increased because a small amount (from 1.5 ppb to 2.5 ppb) of SO$_2$ ended up in the chamber together with the HONO addition. The overall SOA mass increased to 2.2 times the initial the POA mass (Table 3). However, the d-9 butanol decay indicated atmospheric aging equivalent to only 0.4 days (Table 4). This observation may be interpreted by the fact that although the photolysis of HONO acts as the source of OH radicals,

the high concentration of propene and NOx may have scavenged OH radicals, leading to a slower d-9 butanol decay rate than in the other experiments. The UV lights were turned on immediately after ozone injection in experiments 4B and 5B (Fig. 2c). After 4 h of photo-oxidation, the SOA mass increased by factors of 1.8 (Exp. 4B) and 2.1 (Exp. 5B) compared to the initial POA mass. This mass increase occurred quickly; most of the SOA was formed during the first hour of photo-oxidation. The mass increase due to photo-oxidation was 10 µg h$^{-1}$ and 32 µg h$^{-1}$ for fast and slow ignition cases, respectively. The

corresponding first hour dark aging SOA formation rates were 8 µg h$^{-1}$ and 12 µg h$^{-1}$, respectively.

Substantial SOA formation was generally observed in every experiment performed with logwood combustion (birch, beech and spruce). As expected, the VOC-to-NOx ratios affect SOA formation: combustion under higher VOC-to-NOx ratios generates more SOA, showing that the burning conditions are likely a decisive factor in SOA formation. The results also show



that photo-oxidation is faster than dark aging, which is expected. Furthermore, our findings emphasize the importance of dark aging, which appears to generate similar SOA mass loadings as UV aging.

**3.3 OA composition**

The elemental OA composition and atomic ratios were calculated from the high-resolution AMS spectra for each experiment to monitor dynamic changes in the OA composition during aging. The OA spectra of non-aged wood combustion emissions contained high amounts of oxygenated organic species with all wood species. The average carbon oxidation state $OS_C$ of non-aged OA (POA), which can be estimated from using (Kroll et al., 2011; Canagaratna et al., 2015)

$$\overline{OS} \approx 2 \times O:C - H:C, \tag{1}$$

ranged from -0.4 to -0.8 with an average O:C ratio of 0.5 and an H:C ratio of 1.5 (Table 5). The POA oxidation state and OM-to-OC ratio of spruce emissions were higher than their equivalents for beech and birch POA. Higher oxidation states of POA from spruce emissions (Eq. 1) compared to beech and birch emissions can be explained by the softwood material. The organic composition of softwood emissions, such as spruce, are influenced by resin acids, such as pimaric, iso-pimaric, sandaracopimaric and abietic acids (Simoneit and Elias, 2001), that are not emitted from hardwoods such as beech and birch. In addition, the concentrations of aromatic VOCs (Fig. S2) were higher for spruce than for beech and birch. Because these compounds are important precursors of SOA, they are likely to affect the total SOA formation and its characteristics in the spruce combustion experiments.

The $OS_c$ (Fig. 3) increased from approximately $-0.5$ (Table 5) to 0.1–0.3 during dark aging, which is a level of aged secondary organic aerosol that is produced by multiple oxidation reactions (Kroll et al., 2011). An $OS_c$ level of $-0.5$ corresponds to an upper end of fresh biomass burning aerosol, underlining that logwood combustion generates a higher oxidation state OA than wildfire biomass burning. This can be associated with the fact that the emissions source of our study includes the low temperature ignition phase of the wood batch which is shown to emit relatively high amounts of oxidized organics, formed via pyrolytic decomposition of wood polymers (Czech et al., 2016). Moreover, the mass peaks associated with oxygenated OA (OOA), such as $CHO^+$, $C_2H_3O^+$ and $CO_2^+$, increased distinctly during the 4-hour dark aging period and in conjunction with nitrate-related peaks. The highly oxidized ions, i.e., $C_xH_yO_{z>1}^+$ (total of 391 ions), increased more than the ions with intermediate oxygen content, i.e., ($C_xH_yO^+$ (total of 245 ions). Thus, relatively highly oxidized species, such as carboxylic acids or peroxides, dominated the observed increase in the O:C ratio and the oxidation state $OS_c$.

The enhancement in OA oxidation occurred more rapidly during the first 2 hours of dark aging and slowed during the second half of the dark aging period (Fig. 3). Switching on the UV lights after the dark aging period only slightly increased the $OS_c$, although it caused a substantial and rapid SOA mass increase (Figs. 2a and 3a). In contrast, in experiment 3B, oxidation was further accelerated by the injection of HONO immediately before UV light exposure, leading to an $OS_c$ peak value of 0.6 (Fig. 3b), which corresponds to highly oxidized ambient organic aerosol. When the UV lights were switched on directly after the



sample stabilization period (Exp. 4B and 5B), OA oxidation was found to occur quickly (Fig. 3c), although the oxidation state remained at a lower level (from 0 to 0.1) than during the dark aging experiments.

The effect of OH exposure on organic aerosol is summarized in Figure 4, emphasizing the strong effect of dark aging on the OA composition when the oxidation state increased strongly without UV aging. Fast ignition (Exp. 1B and 4B) generated more oxidized OA than slow ignition (Exp. 2B and 5B), although these differences decreased when the SOA mass fraction increased. An intermediate slope of $-1$ in Van Krevelen diagrams (Heald et al., 2010; Fig. 4c) corresponds to an addition of carboxylic acid groups without fragmentation, which would increase the slope toward $-0.5$ (Ng. et al., 2011). Figure 4 depicts Van Krevelen slope values of $-0.64$ to $-0.67$ for the experiments; these slopes were found to be slightly steeper for the fast ignition experiments 1B and 4B than for the dark aging experiments 2B and 5B. These slope values range in Van Krevelen space over a region where ambient OA components appear (Ortega et al., 2015). The OA aging was found to follow a similar trend with ambient observations (Ng et al., 2011), showing that our chamber experiments simulate polluted boundary-layer conditions. The effect of HONO injection on the OA composition can be observed as a Van Krevelen slope increase. This change in the slope can be explained by the replacement of a hydrogen with an alcohol group (-OH).

## 3.4 SOA composition

### 3.4.1 PMF analyses of the second set of experiments

To investigate the OA composition in the detail, positive matrix factorization (PMF) was used to separate SOA, POA, and their subgroups from the total OA. The method of analyzing AMS data with PMF is described in Lanz et al. (2007) and Ulbrich et al. (2009). In addition to OA spectra, inorganic NO and $NO_2$ spectra were added to the PMF to investigate particulate organic nitrate formation following Hao et al., 2014 and Kortelainen et al. 2016. To date, most AMS-PMF analyses have been performed with ambient measurements (e.g., Zhang et al., 2011), although laboratory-based AMS datasets have been successfully analyzed using this method in recent years (e.g., Kortelainen et al., 2015; Corbin et al., 2015b).

PMF was conducted with a five-factor solution. Solutions with more factors do not increase the physical realistic meaning; for example, no correlation with tracers can be identified. Thus, a five-factor solution was chosen for further investigation (Supplement S1). The factor identification was confirmed by comparing the time series and mass spectra of each factor with external particulate tracers ($NO_3^-$, $SO_4^{2-}$, $NH_4^+$, $Cl^-$, PAH, $C_2H_3O^+$ and $C_4H_9^+$), available gas-phase measurements (NO, $NO_2$, $NO_x$, $SO_2$ and $O_3$) (Fig. S5), reference source mass spectra available from the AMS MS database (Lechner, M., Ulbrich, I.M., and Jimenez, J.L. High-Resolution AMS Spectral Database. URL: http://cires.colorado.edu/jimenez-group/HRAMSsd/; Ulbrich et al., 2009), and wood combustion mass spectra from Kortelainen et al. 2015 and Aiken et al. 2009 (Table S2). The following factors were identified: POA1, POA2, SOA1, SOA2 and SOA3 (Table 6).



### 3.4.2 Interpretation of the PMF results

Figure 5 illustrates the evolution of the OA composition based on the PMF analysis. The aging processes of organic aerosol include functionalization reactions (the addition of oxygen-containing functional groups), which continuously decrease the volatility of organic compounds, and fragmentation reactions (the breaking of carbon-carbon bonds), which simultaneously result in higher-volatility products (Kroll et al., 2011; Lambe et al., 2012). Oligomerization, polymerization, and heterogeneous oxidation typically appear on longer time scales; thus, we do not expect these mechanisms to significantly affect SOA aging when chemical aging times are relatively short, i.e., less than 24 h.

PMF differentiates three SOA factors, SOA1, SOA2, and SOA3 (Fig. 5 and Fig. S5), representing the three major oxidizers: ozone, the OH radical, and the nitrate radical (e.g., Kroll and Seinfeld 2008; Chacon-Madrid and Donahue 2011; Rollins et al., 2012). During dark aging, when the OH concentrations were low, ozonolysis and nitrate radicals dominated the oxidation processes of VOCs and subsequent formation of SOA. In contrast, when the UV lights were switched on, OH radicals dominated the aging process. The species in the first SOA factor (SOA1) are expected to be formed via ozonolysis of unsaturated compounds, e.g., alkenes, dienes, terpenes, etc., because other major compound classes are not sufficiently reactive with ozone. This can be seen in the SOA1 factor, which was higher in the slow ignition case with higher alkene fractions than in the fast ignition case (Fig. S2) when SOA1 was only less than ~10% of the total SOA. The oxidation level of SOA1 was clearly lower than SOA2 and SOA3 (Table 6), suggesting that the oxidation products of ozonolysis from logwood combustion are more volatile than the oxidation products of $NO_3$ or OH radicals because ozone only attacks double-bond VOCs (e.g., Chhabra et al., 2010).

The species in the second SOA factor (SOA2) are most likely present as secondary organonitrates (ON), which is indicated by the following observations. First, the SOA2 factor exhibited a similar time series to that of particulate $NO_3$ (Fig. S5), and there was a lack of base, such as $NH_3$, to form inorganic nitrate. Second, SOA2 increased considerably during dark aging and declined during UV-induced aging (Fig. 5). This agrees with the earlier findings on organonitrate formation in ambient air (Rollins et al., 2012; Fry et al., 2013; Kortelainen et al., 2016; Kiender-Scharr et al., 2016) and with the high $NO$-to-$NO_2$ ratio of the SOA2 factor. Third, the mass spectrum of SOA2 has $NO$-to-$NO_2$ ion signal ratio of 7.8, which is distinctly higher than the corresponding ratio of 2.5 in the standard ammonium nitrate (AN) calibration. Such a $NO$-to-$NO2$ ratio indicates the presence of organonitrates (Farmer et al., 2010; Kiendler-Scharr et al., 2016). This result is further indicated by the particle size distributions for $NO_3$ and OA being identical. The SOA2 concentration increased quickly during the first hour of dark aging and reached a plateau before the UV aging period. SOA2 decreased after the UV lights were switched on to a similar or slightly higher level than during the stabilization phase. This result can be explained by the decomposition of organonitrates under UV irradiation (e.g., Nguyen et al., 2015; Kortelainen et al., 2016).

The third SOA factor (SOA3) was primarily formed after the UV lights were switched on; it represents SOA formation dominated by the OH radical. SOA3 exhibited the highest oxidation state, 0.6 (Table 6), among the identified SOA factors,



comparable to a level of highly oxidized ambient OA (Kroll et al., 2011). The potential precursors for this factor are likely aromatic compounds (e.g., toluene and xylene), which have typically slow reaction rates in the presence of ozone. This conclusion is indicated by the clearly higher SOA3 mass in the slow ignition case, which contained higher amounts of aromatic compounds in the primary emissions, than in the fast ignition case (Fig. S2). After UV aging, SOA3 represented about half of the total SOA mass for fast ignition (Exp. 1B and 4B) and 70% for slow ignition (Exp. 2B and 5B). The PMF results also revealed that after HONO addition and subsequent UV aging, the SOA3 factor represented 94% of the SOA mass. Small fractions of SOA3 appeared during dark aging, which is a potential indication of OH production by ozone without UV lights when a criegee is falling apart (Table 4, Seinfeld and Pandis, 2006).

POA1 (BBOA factor including PAH) was higher during the slow ignition cases than the fast ignition experiments and decreased sharply after the UV lights were switched on because of the degradation of PAH by OH (e.g., Keyte et al., 2013), whereas UV aging did not have a similarly well-defined influence on POA2 (HOA factor). The majority of PAH compounds are semi-volatile and thus partition between the vapor and condensed phases. Both the vapor and particulate forms undergo chemical reactions, as observed by the decrease in POA1. The PAH concentrations, which ranged from 0.1 to 0.5 µg m$^{-3}$ (Fig. S5) (about 1 % of OA), were estimated based on the UMR method of Dzepina et al. (2007). The method has some limitations, e.g., interference by non-PAH compounds at higher m/z, which could explain differences between PAH and POA1 during slow ignition UV aging when the non-PAH concentrations were high (Fig. S5a).

POA was found to oxidize after the ozone addition, forming aged POA (Hennigan et al., 2011), which can be defined as the concentration of the initial POA minus the residual POA, based on the PMF calculations, at the end of the experiment (Figs. 5 and 6). The non-aged POA mass fraction (Exp. 1B-3B) was approximately 40−50% after dark aging and decreased to 8-23% after UV aging. UV aging experiments 4B and 5B without dark aging showed higher non-aged POA fractions of 54% and 27% for fast and slow ignition, respectively. These results are quite similar to Hennigan et al. (2011) for the photo-oxidation of open biomass burning experiments. They concluded that unreacted POA contributed 17% of the campaign-averaged OA mass, and heterogeneous reactions with OH could account for less than half of this POA transformation. The transformation of semi-volatile OA to low-volatile OA can involve both gas-phase oxidation followed by partitioning to the particle phase as well as heterogeneous reactions. Recent studies have shown that the reaction of OH or O$_3$ radicals with the SOA coating is a potential oxidation path (Browne et al., 2015). Liu et al. (2012) showed that heterogeneous oxidation of PAHs driven by NO$_3$ radicals can be faster than their corresponding gas-phase reactions when OH radical concentrations are low, i.e., during dark aging.

### 3.4.3 Organonitrates

Organonitrate (ON) mass can be calculated from (Farmer et al., 2010; Fry et al., 2013; Kiendler-Scharr et al., 2016)

$$OrgNO3_{frac} = \frac{(1+R_{OrgNO3}) \times (R_{measured} - R_{calib})}{1+R_{measured}) \times (R_{OrgNO3} - R_{calib})} \tag{2}$$



$$OrgNO3_{mass} = OrgNO3_{frac} \times NO_3 \tag{3}$$

where $R_{measured}$ is the measured intensity ratio of $NO_2$ and $NO$ ions as a function of time in the individual data sets, $R_{calib}$ is the ratio observed in the $NH_4NO_3$ calibrations (1.7 for Exp.1A-3A and 2.5 for Exp. 1B-5B), and $R_{OrgNO3}$ is set to 0.1 (Kiendler-Scharr et al., 2016). This expression only applies if $NH_4NO_3$ is the only important inorganic nitrate addition to ON in the

submicron mode because other nitrate salts have different fragmentation ratios (Farmer et al., 2010; Fry et al., 2013). However, alkali metals emitted from wood combustion, as mentioned in Section 3.1, appear mainly in the form of $A_2SO_4$, $ACl$ and $A_2CO_3$ (A = K or Na) (Sippula et al., 2007). Furthermore, K and Na as well as $CO_3$ typically have relatively low concentrations in logwood combustion primary emissions (Nuutinen et al., 2014), which are clearly below the concentrations of nitrate in this study. Thus, it is impossible that alkali metal ions contributed significantly to the submicron ON during our experiments.

The mass spectra of OA (Fig. S5) suggested that, in addition to SOA2, NO and $NO_2$ ions were also present as primary factors (POA1 and POA2) and a secondary factor (SOA3). The total ON mass was calculated using Eq. (2) and (3) with corresponding emissions of 48 – 75 mg kg$^{-1}$ fuel (Table 7). Nitrate radical reactions and other mechanisms, such as termination reactions or RO2 with NO to ROONO and rearrangement to RONO2, could be the production mechanism of organonitrates.

## 4 Conclusions

Real-time measurements of OA aging and SOA formation from logwood combustion were conducted in a 29 m$^3$ smog chamber under dark (i.e., $O_3$ and $NO_3$) and UV (i.e., OH) oxidation environments. The results represent modern masonry heater emissions used in northern Europe. Substantial SOA formation was observed in all dark and UV aging experiments, leading to nearly twice the initial OA mass. Higher SOA mass was observed for slow ignition (VOC to NO$_X$ ratio = 5) than for fast ignition (VOC to NO$_X$ ratio = 3), which emphasizes the importance of the
burning conditions for the aging processes.

A high fraction of organonitrates (ON) was emitted from the combustion of all wood species, whereas secondary ON was only formed during dark aging. Secondary ON were formed from the oxidation of VOCs by $NO_3$ radicals under nocturnal conditions and in the presence of ozone. These results imply that wood combustion is a significant source of ON to the atmospheric boundary layer.

Most of the POA was found to become oxidized after the ozone addition, forming aged POA, such that the remaining non-aged POA represented less than 25% of the total POA after 7 h of aging. It is likely that the aging process induces changes in the semi-volatile organic compounds in the POA such as PAH and oxy-PAH, which can impact the toxicity. The observed transformation of POA should be further evaluated and included in emission inventories and global model calculations.

The results obtained from this investigation provide new insights into SOA formation and transformation processes in polluted
atmospheric conditions (plumes). These are prevalent, e.g., in the commonly observed haze pollution events in China, whose



features are very different from those reported for non-polluted aerosol systems in the atmosphere. Furthermore, the SOA formation from dark aging and aging by photochemical reactions were connected for the first time to examine the fate of SOA from nighttime exposed to UV light. The results generally prove that logwood burning emissions are subject to intensive chemical processing in the atmosphere, with time scales of the order of hours including dark periods.

## 5 Data availability

The data from this paper can be obtained by contacting the authors of this article.

### Acknowledgments

Financial support by the Academy of Finland (Grant: 258315), HICE Helmholtz Virtual Institute, the strategic funding of the University of Eastern Finland for the project on sustainable bioenergy, climate change and health, the DACH project WOOSHI
and the Maj and Tor Nessling Foundation (201600029) are acknowledged.

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





Table 1: Overview of the experimental parameters in the smog chamber experiments: VOC-to-NOx ratio (VOC/$NO_x$) (ppbv/ppbv), non-methane VOC-to-NOx ratio (NMVOC/$NO_x$) (ppbv/ppbv), dilution ratio (DR) and the specific emission conversion factors (ECFs) ($m^3$ $kg^{-1}$ fuel) determined for each logwood combustion experiment. The ECF calculations followed Reda et al. (2015).

| Exp. | Wood | Function | Ignition | VOC/ $NO_X$ | NMVOC/ $NO_X$ | DR | ECF[a] |
|---|---|---|---|---|---|---|---|
| 1A | Beech | Dark | Fast | 2.6 | 1.2 | 372 | 7.8 |
| 2A | Birch | Dark | Fast | 1.3 | 0.6 | 456 | 9.1 |
| 3A | Spruce | Dark | Fast | 2.7 | 1.2 | 313 | 6.9 |
| 1B | Spruce | Dark + UV | Fast | 3.1 | 1.3 | 259 | 5.8 |
| 2B | Spruce | Dark + UV | Slow | 4.9 | 2.3 | 245 | 6.2 |
| 3B | Spruce | Dark + UV + H[b] | Fast | 4.0 | 1.5 | 245 | 6.0 |
| 4B | Spruce | UV | Fast | 2.7 | 1.1 | 266 | 6.1 |
| 5B | Spruce | UV | Slow | 4.8 | 2.2 | 229 | 5.8 |

5    [a] The ECF calculations followed Reda et al. (2015.
[b] HONO addition





Table 2: PM emissions (mg kg$^{-1}$ fuel) of the AMS species and ratios of OA to rBC and PM (TEOM) to AMS total.

| Exp. | OA | rBC | NO$_3$ | SO$_4$ | NH$_4$ | Chl | Total | OA/rBC | PM/Total |
|------|------|------|------|------|------|------|-------|--------|----------|
| 1A | 209.8 | 751.1 | 40.8 | 27 | 0.5 | 0.9 | 1029 | 0.28 | - |
| 2A | 221.2 | 1133 | 37.1 | 8.8 | 0.2 | 2.0 | 1403 | 0.20 | - |
| 3A | 105.9 | 512.2 | 22.5 | 6.7 | 0.8 | 1.5 | 649.6 | 0.21 | - |
| 1B | 157.7 | 513.7 | 42.4 | 8.7 | 0.6 | 0.6 | 723.7 | 0.31 | 1.2 |
| 2B | 228.3 | 655.5 | 43.0 | 5.9 | 0.6 | 0.7 | 934.0 | 0.35 | 1.3 |
| 3B | 149.5 | 572.2 | 34.1 | 6.3 | 0.4 | 0.7 | 763.1 | 0.26 | 1.2 |
| 4B | 118.3 | 422.0 | 42.7 | 6.7 | 1.2 | 0.7 | 591.6 | 0.28 | 1.3 |
| 5B | 207.1 | 546.0 | 35.9 | 7.4 | 0.3 | 1.1 | 797.9 | 0.38 | 1.3 |





Table 3: SOA (OA-POA) emissions (mg kg$^{-1}$ fuel) and SOA-to-POA/ OA-to-POA ratios for all of the experiments. Dark aging results are shown in bold.

| Exp. | Dark/UV aging | | | Dark + UV aging | | |
|---|---|---|---|---|---|---|
| | SOA | SOA/POA | OA/POA | SOA | SOA/POA | OA/POA |
| 1A[a] | **126** | **60 %** | **1.6** | - | - | - |
| 2A[a] | **170** | **77 %** | **1.8** | - | - | - |
| 3A[a] | **175** | **165 %** | **2.6** | - | - | - |
| 1B | **100** | **63 %** | **1.6** | 137 | 87 % | 1.9 |
| 2B | **172** | **75 %** | **1.8** | 308 | 135 % | 2.4 |
| 3B | **101** | **68 %** | **1.7** | 173 | 116 % | 2.2 |
| 4B[b] | 91 | 77 % | 1.8 | - | - | - |
| 5B[b] | 237 | 115 % | 2.1 | - | - | - |

[a] Only dark aging
[b] Only UV aging



Table 4: Table 4: Chemical ages (days) for the second set of chamber experiments (Exp. 1B-5B) and corresponding parameters: average OH concentration (molec. cm$^{-3}$) and OH exposure (molec. cm$^{-3}$ s) based on the decay of butanol-d9 (Barmet et al., 2012). The equivalent chemical age was calculated assuming an average atmospheric OH concentration of $1.0 \times 10^6$ molecules cm$^{-3}$.

| Exp. | Dark aging | | | UV aging | | |
| | OH conc. | OH exp | Age | OH conc. | OH exp. | Age |
|---|---|---|---|---|---|---|
| 1B | - | - | - | 4.6E+6 | 5.0E+10 | 0.58 |
| 2B | 6.6E+5 | 9.2E+9 | 0.11 | 4.9E+6 | 5.2E+10 | 0.59 |
| 3B | 4.7E+5 | 7.7E+9 | 0.10 | 2.4E+6 | 2.6E+10 | 0.30 |
| 4B | - | - | - | 3.4E+6 | 5.1E+10 | 0.58 |
| 5B | - | - | - | 4.6E+6 | 6.6E+10 | 0.77 |



Table 5: Overview of the OA elemental analysis at the end of each experimental stage (stabilization, dark aging, and UV aging); dark aging results are shown in bold.

| Exp. | O:C | H:C | OM:OC | OS | Dark/UV aging O:C | H:C | OM:OC | OS | Total O:C | H:C | OM:OC | OS |
|------|-----|-----|-------|-----|------|-----|-------|-----|------|-----|-------|-----|
| 1A | 0.41 | 1.55 | 1.69 | -0.73 | **0.83** | **1.31** | **2.21** | **0.35** | - | - | - | - |
| 2A | 0.35 | 1.50 | 1.60 | -0.80 | **0.75** | **1.29** | **2.11** | **0.21** | - | - | - | - |
| 3A | 0.53 | 1.48 | 1.84 | -0.42 | **0.77** | **1.43** | **2.15** | **0.11** | - | - | - | - |
| 1B | 0.55 | 1.52 | 1.86 | -0.42 | **0.81** | **1.33** | **2.19** | **0.29** | 0.83 | 1.31 | 2.21 | 0.34 |
| 2B | 0.53 | 1.50 | 1.84 | -0.44 | **0.74** | **1.35** | **2.10** | **0.13** | 0.82 | 1.30 | 2.20 | 0.33 |
| 3B | 0.47 | 1.56 | 1.75 | -0.62 | **0.78** | **1.33** | **2.15** | **0.22** | 0.96 | 1.29 | 2.39 | 0.63 |
| 4B | 0.55 | 1.51 | 1.87 | -0.41 | 0.73 | 1.38 | 2.09 | 0.08 | - | - | - | - |
| 5B | 0.51 | 1.52 | 1.81 | -0.50 | 0.75 | 1.36 | 2.11 | 0.14 | - | - | - | - |



Table 6: Elemental composition of the PMF factors POA1, POA2, SOA1, SOA2 and SOA3.

| Factor | O:C | H:C | OM:OC | OS |
|--------|-----|-----|-------|-----|
| POA1 | 0.46 | 1.58 | 1.74 | -0.67 |
| POA2 | 0.45 | 1.58 | 1.74 | -0.69 |
| SOA1 | 0.60 | 1.40 | 1.92 | -0.20 |
| SOA2 | 0.86 | 1.26 | 2.26 | 0.47 |
| SOA3 | 0.95 | 1.28 | 2.38 | 0.62 |





Table 7: Organonitrate (ON) emissions (mg kg$^{-1}$ fuel) from logwood combustion at the end of experimental periods (stabilization and dark aging) calculated using Eq. (2) and (3) and corresponding NO-to-NO2 ratios.

| Exp. | Stabilization | | Dark aging | | Total |
|------|--------------|----------|-----------|----------|-------|
|      | NO/NO$_2$ | Emission | NO/NO$_2$ | Emission |       |
| 1A | 5.6 | 31.2 | 5.9 | 24.6 | 55.8 |
| 2A | 5.3 | 27.4 | 8.3 | 47.7 | 75.1 |
| 3A | 10.0 | 22.5 | 10.0 | 28.5 | 51.0 |
| 1B | 6.9 | 34.9 | 7.9 | 21.5 | 56.4 |
| 2B | 5.1 | 26.7 | 5.8 | 24.3 | 51.0 |
| 3B | 5.4 | 22.7 | 6.5 | 25.7 | 48.4 |
| 4B | 5.5 | 29.3 | - | - | 29.3 |
| 5B | 4.5 | 19.8 | - | - | 19.8 |





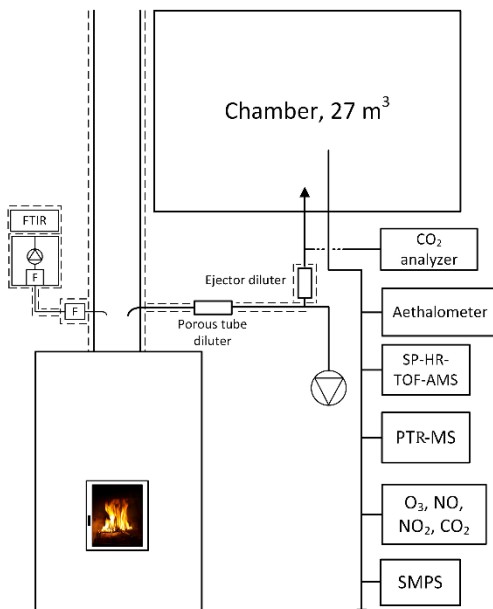

**Figure 1: Experimental setup in the ILMARI combustion research facility at the University of Eastern Finland.**



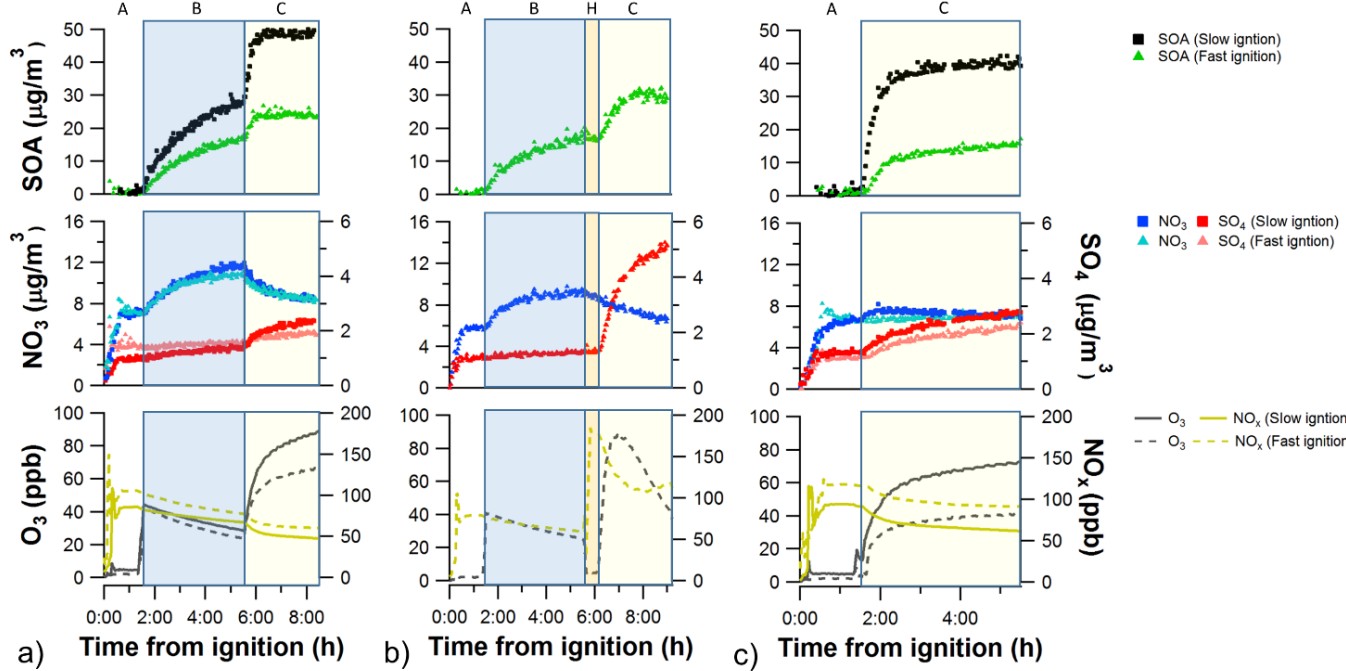

**Figure 2: SOA mass and selected trace gas concentrations in a smog chamber for experiments (a) 1B and 2B, (b) 3B, and (c) 4B and 5B. Stabilization (40 min) followed by filling the chamber (35 min) is marked as A, dark aging as B, UV aging as C, and HONO addition as H. $NO_x/5$ and $O_3/7$ after the HONO addition is shown in Fig 2b.**





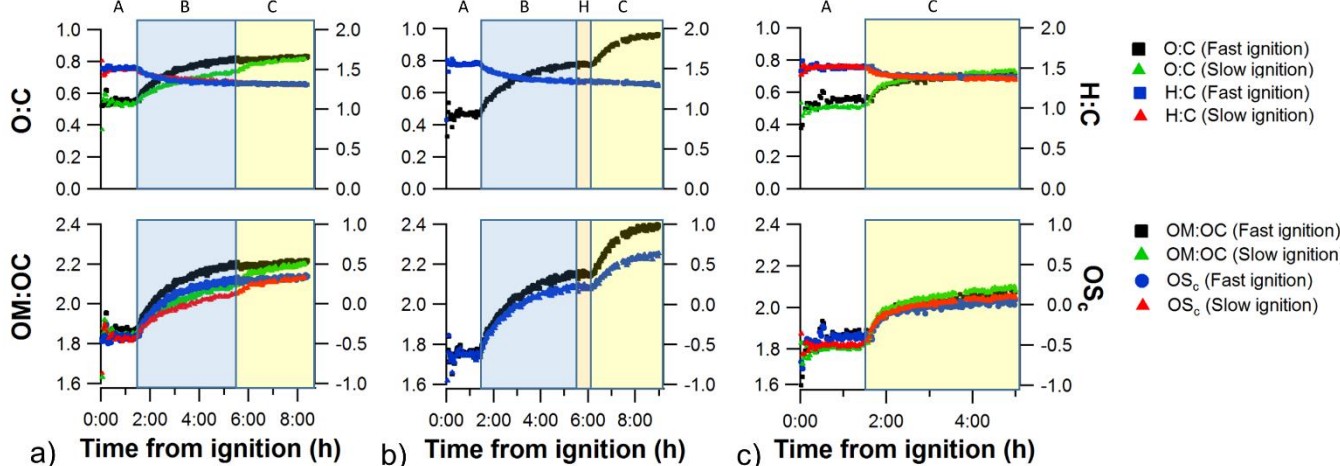

**Figure 3: Effect of aging on OA elemental composition for experiments (a) 1B and 2B, (b) 3B and (c) 4B and 5B.**







**Figure 4: Comparison of OA elemental composition for the second set of experiments (Exp. 1B−5B): OA elemental composition vs. OH exposure (a-b) and Van Krevelen diagram (c). The improved-ambient method from Canagaratna et al. (2015) was applied for the elemental analysis. The dotted lines define the triangular space where ambient OA components are typically located (Ng et al., 2010) and dark aging periods are circled in grey (a-b).**



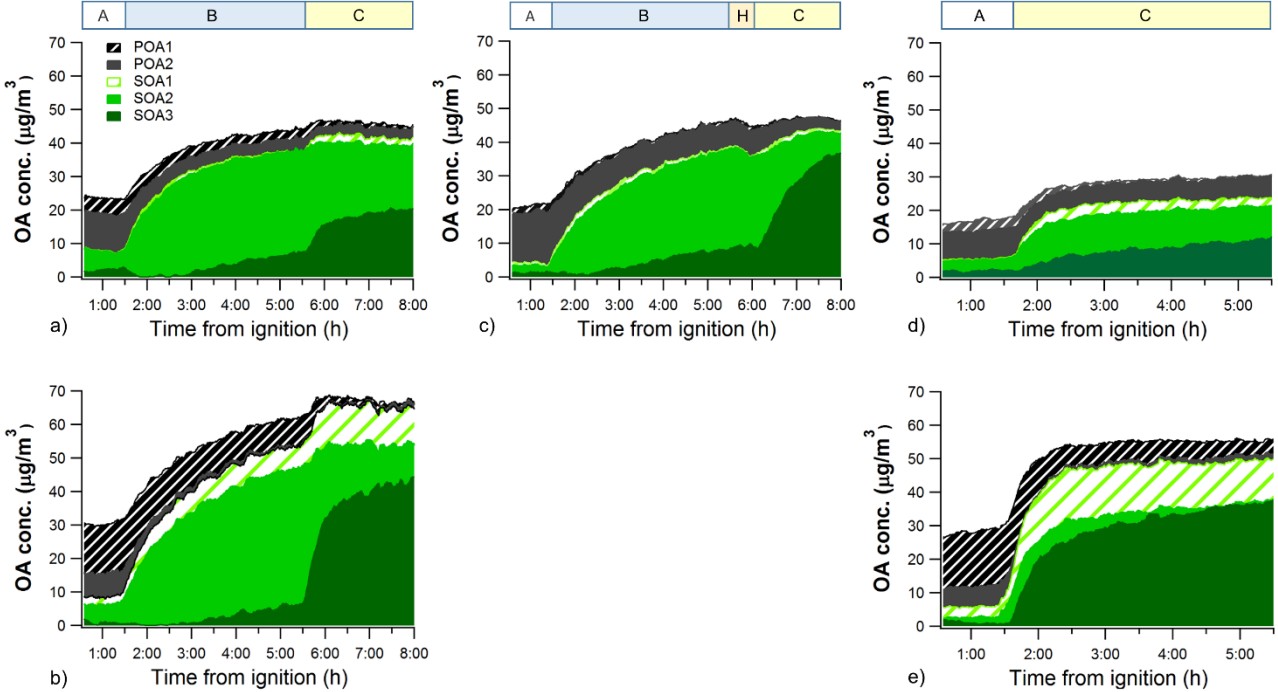

**Figure 5: Evolution of OA during aging based on PMF calculations: (a) Exp. 1B, (b) Exp. 2B, (c) Exp. 3B, (d) Exp. 4B, and (e) Exp. 5B. The stabilization phase is marked as A, dark aging as B, UV aging as C, and the HONO addition as H in the upper panel.**



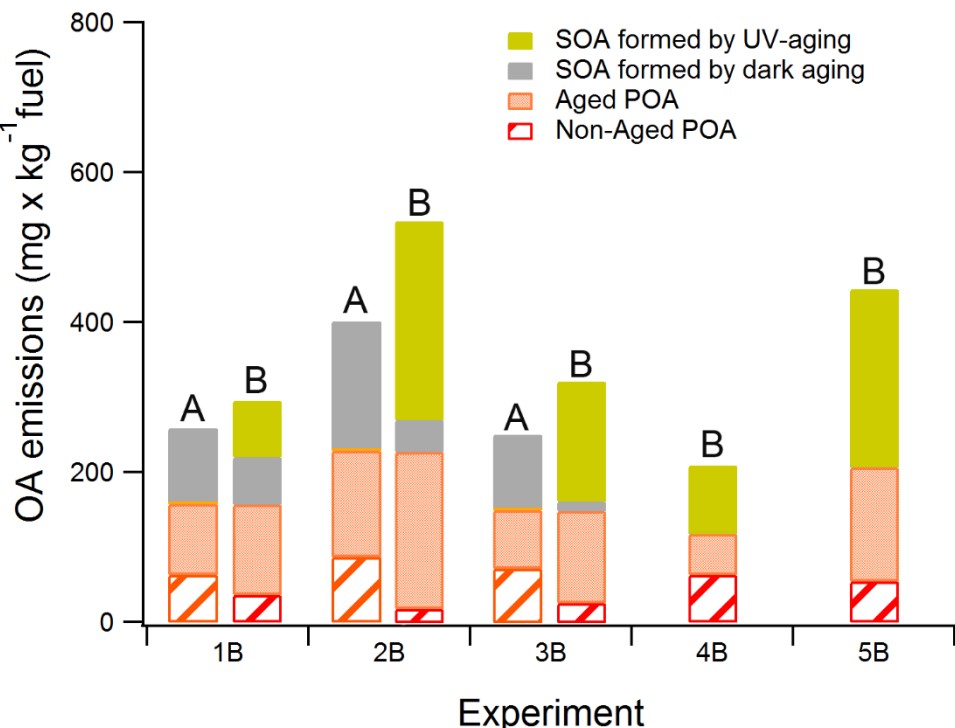

**Figure 6: Aged and non-aged POA emission together with SOA at the end of dark (A) and UV aging (B) periods based on PMF calculations (Exp. 1B-5B).**