# Peer review of "Transformation of logwood combustion emissions in a smog chamber: formation of secondary organic aerosol and changes in the primary organic aerosol upon daytime and nighttime aging"

_Atmospheric Chemistry and Physics, 2016_

## Referee Comment (RC1) · Anonymous Referee #1 · 2 Jul 2016

The manuscript by Tiitta et al. "Transformation of logwood combustion emissions in a smog chamber: formation of secondary organic aerosol and changes in the primary organic aerosol upon daytime and nighttime aging" is important since 1) there is a lack of knowledge on the SOA formation from biomass emissions and 2) biomass combustion emissions are extremely heterogeneous in their properties. A highlight is the importance of dark aging on POA transformation and SOA formation and the quantification of fresh and aged organic nitrates. The use of PMF is an important tool that here is exploited to a more sophisticated degree than in previous papers on biomass SOA. An impressive array of highly suitable scientific methods have been used in the study and

the presentation is scientifically sound. The paper is suitable for publication in ACP after major revision according to the points below.

Abstract: At 496 words the abstract is not optimized to capture the interested reader. I would suggest to reduce the abstract to 300 words to more efficiently convey the main findings.

P4:R28-: The authors chose to investigate cold start (kindling) emissions using a very dry fuel. Since this a very important study the reader needs to know about the generalizability of the chosen combustion variables. How would the findings transform to fuel addition on glowing embers (as I would suspect is more representative for ambient biomass PM) and fuel with higher moisture content?

For example P6R4: The authors make several notes about the relevance for Chinese conditions. I agree biomass combustion emissions are very important for air quality in China. However, for the logic to make sense please comment in the paper if the chosen combustion system (and fuels) are of relevance for Chinese biomass combustion emissions.

P6R30: Dual or Tungsten vaporizer? Which was used in the presented results beyond the rBC data? Please state. Please check that the 60 s is enough to get a stable signal without being sensitive to transients when laser goes on-off. Was there any transients observed or were the results "perfect square waves"?

P7R8: As I understand it a reduced CE (0.6) was needed compared to the mIE calibration with Regal Black (RB). Thus the sensitivity of the SP-AMS to biomass rBC was lower than for RB. Note that this is opposite to the results by Willis et al. (2014) who found that the sensitivity increased upon coating RB. I do not argue against the CE of 0.6 for biomass compared to RB, it is similar to published data for wood stoves (Martinsson et al. 2015). However, it should be motivated based on the TEOM comparison or biomass data in literature and not based on the Willis et al. reference.

[Figure]

P7R21: So the CE of rBC did not change much upon aging? Please clearly state if this is correct or not. I assume it makes sense since the rBC particles are only thinly coated by aging.

P9R16: It is very good that it is pointed out that the SP-AMS measurement of K is highly uncertain. However, also the quantification of Cl and SO4 requires caution as their chemical state is quite different from ambient observations. Their quantification from SP-AMS data is new for me in biomass combustion emissions. Please add to the methods section how this was done (vaporizer mode, CE, RIE etc). As is stated K salts may be present as external mixtures. If so what would make the rBC free particles vaporize in the W and dual vaporization modes respectively, given their absorption cross sections and vapor pressures? Was there any filter samples to validate the new method to quantify Chl and SO4 in biomass combustion data?

Figure 2: The small and capital letters a, b, c risks the leader will find it hard to quickly understand the figure. Please change the denotation for the different stages of each experiment to something that is more helpful to the reader.

Figure 2b: If the SO4 increase after HONO addition is most likely an artefact then this should be mentioned in the figure caption. However, it is very good that it is mentioned and not sorted away as may sometimes be the case.

Table 4: double legend, please remove

Table 4: It is very good the age is calculated for OH exposure. Could relevant numbers be derived also for the dark aging?

P10R26: If ozone was added in the "UV-only" experiments, could you then conclude that OH chemistry and not O3 chemistry is responsible for the SOA formation in this case?

P11R5: Was the laser vaporizer engaged when calculating the elemental ratios? Please describe in the text. This is critical since the laser may produce a relatively

strong CO2+ signal which have been hypothesized to origin from surface oxides on the soot (Corbin et al. 2015). It is not clear to me if this signal should be attributed to the rBC core or the OA coating. Thus, one may speculate that the O:C would be significantly higher for dual vaporisers than W-vaporiser. Please comment in the text the O:C for fresh emissions with dual and W vaporizer, respectively.

Figure 5: Please change the names of the POA and SOA factors to represent what they are tracing, it would make it much easier to convey the messages to the reader. Add info about fast and slow ignition to the figure, this is currently missing.

P14R2: Yes possibly toluene, but Xylenes commonly have a much higher f43/f44 ratio than found here. What about Naphthalene? Another group of important SOA-precursors are phenols. Please comment about their potential contribution to the different factors based on their reactivity towards different oxidants. An important paper on the precursors responsible for SOA formation was recently published by Bruns et al. (2016). Please add this reference and compare your conclusions with their findings.

Figure 6: It would be important to include the rBC emission factors in this figure (on a separate axis) so it becomes clear for the reader that these aerosols are always rBC dominated.

P14R9: POA seems to contain very low PAH contribution, why would then PAH degradation by UV be a major course of the reduction of this factor with UV?

References:

Bruns, Emily A., et al. "Identification of significant precursor gases of secondary organic aerosols from residential wood combustion." Scientific Reports 6 (2016).

Corbin, J. C., et al. "Black carbon surface oxidation and organic composition of beech-wood soot aerosols." Atmospheric Chemistry and Physics 15.20 (2015): 11885-11907.

Martinsson, Johan, et al. "Impacts of combustion conditions and photochemical processing on the light absorption of biomass combustion aerosol." Environmental science

& technology 49.24 (2015): 14663-14671.

Willis, M. D., Lee, A. K. Y., Onasch, T. B., Fortner, E. C., Williams, L. R., Lambe, A. T., ... & Abbatt, J. P. D. (2014). Collection efficiency of the Soot-Particle Aerosol Mass Spectrometer (SP-AMS) for internally mixed particulate black carbon. Atmospheric Measurement Techniques, 7(12), 4507-4516.

———————————————————

---

## Referee Comment (RC2) · Anonymous Referee #2 · 22 Jul 2016

The manuscript by Tiitta et al., is a very good piece of work presenting important findings on the dark and UV ageing of emissions from small-scale log wood combustion. The study investigates the ageing of primary emissions and the formation of secondary particulate matter using a state-of-the-art experimental facilities and measurement and analytical techniques. The paper is certainly within the remit of ACP and a revised manuscript should be considered for publication after the following comments are addressed:

A. General comments:

[Figure]

1. Given that a substantial amount of discussion in the manuscript is made within the context of fast vs slow ignition, the authors should dedicate a paragraph in the results section to discuss the main differences between the two methods in terms of emission profiles in order to help and guide the reader throughout the discussion.

2. PMF Factors: The SOA2 factor suggested to be a result of NO3 chemistry should only be important during dark ageing experiment. The fact that the concentration of this factor remain high (or even slightly increase) in Figure 5 suggest that other oxidation sources are important to its formation or that the light source in the chamber does fully represent day time chemistry in the troposphere. This should be discussed and clarified in the revised manuscript. On a related note, the correlation between SOA2 and NO3 mass in Figure S5 should be supplemented by more information showing (perhaps a lack of) relationship between nitrate and SOA1 and SOA3 in order to confirm the argument suggested by the authors.

B, Minor comments:

1. Page 1, line 21-22: statement about substantial contribution from SOA from small scale wood combustion to global atmospheric PM matter needs to be supported and referenced in the introduction. This issue is more likely to have local/regional impact. In either case, references should be added to support a correct statement.

2. Page 1, line 24: replace "deployed" with "used", "employed" or "utilised"

3. Page 2, line 21: the phrase "and internally mixed" should be removed from this sentence. The mixing state of these components is not universal and it is known to change depending on source and conditions.

4. Page 3, line 13: change "generateproducts" to "generate products"

5. Page 6, line 3-4: What is the relevance of the extra statement on OH concentrations in China to this study?

6. Page 7, line 15: References should be made to two very recent and important

studies in this area by (Ye et al., 2016 and Krechmer et al., 2016).

7. Page 10, line 21-22: The authors should comment on whether or not the small increase in SO2 from 1.5ppb to 2.5ppb from HONO addition is sufficient to explain the increase in SO4 particle mass from about 1 to 5 ug/m3 shown in Figure 2b?. A comment about the nature of the sulphate particles should be made given the remark made in the manuscript about a "lack of base such as NH3" (page 13, line 21).

8. Page 13, line 16-18: The direct connection made by the authors between the "level of oxidation" and "volatility" of the different types of SOA is not really supported by any volatility measurements in this work. Although it is true that other papers in the literature have made such association for other types of particles (e.g. ambient), it is advised that these statements are either supported directly by measurements or to be appropriately toned down.

9. Page 14, line 9-10: The comment regarding a sharp decrease in POA1 after the UV lights were switched on is not consistent with data shown in Figure 5; none of the plots in this figure show a sharp decrease in POA1. This should be clarified and revised.

References: Krechmer, J. E., Pagonis, D., Ziemann, P. J., and Jimenez, J. L.: Quantification of Gas-Wall Partitioning in Teflon Environmental Chambers Using Rapid Bursts of Low-Volatility Oxidized Species Generated in Situ, Environmental Science & Technology, 50, 5757-5765, 10.1021/acs.est.6b00606, 2016.

Ye, P., Ding, X., Hakala, J., Hofbauer, V., Robinson, E. S., and Donahue, N. M.: Vapor wall loss of semi-volatile organic compounds in a Teflon chamber, Aerosol Science and Technology, 50, 822-834, 10.1080/02786826.2016.1195905, 2016.

---

## Author Comment (AC1) · 2 Sep 2016

We thank the referee 1 for a comprehensive review of our manuscript. The referee comments were valuable and we believe that addressing the issues raised by the referee has considerably improved our manuscript.

Comment 1, Abstract: At 496 words the abstract is not optimized to capture the interested reader. I would suggest to reduce the abstract to 300 words to more efficiently convey the main findings.

Answer 1: We agree with the reviewer and reduced the abstract to 336 words as fol-

lows:

"Abstract. Organic aerosols (OA) derived from small-scale wood combustion emissions are not well represented by current emissions inventories and models, although they contribute substantially to the atmospheric particulate matter (PM) levels. In this work, a 29 m3 smog chamber in the ILMARI facility of the University of Eastern Finland was utilized to investigate the formation of secondary organic aerosol (SOA) from a small-scale modern masonry heater commonly used in northern Europe. Emissions were oxidatively aged in the smog chamber for a variety of dark (i.e., O3 and NO3) and UV (i.e., OH) conditions, with OH concentration levels of $(0.5–5)\times 106$ molecules cm-3, achieving equivalent atmospheric aging of up to 18 hours. An aerosol mass spectrometer characterized the direct OA emissions and the SOA formed from the combustion of three wood species (birch, beech and spruce) using two ignition processes (fast ignition with a VOC-to-NOx ratio of 3 and slow ignition with a ratio of 5). Dark and UV aging increased the SOA mass fraction with average SOA productions 2.1 times the initial OA mass loadings. SOA enhancement was found to be higher for the slow ignition compared with fast ignition conditions. Positive matrix factorization (PMF) was used to separate SOA, POA, and their subgroups from the total OA mass spectra. PMF analysis identified two POA and three SOA factors that correlated with the three major oxidizers: ozone, the nitrate radical and the OH radical. Organonitrates (ON) were observed to be emitted directly from the wood combustion and during oxidation via NO3 radicals (dark aging), suggesting small-scale wood combustion may be a significant ON source. POA was oxidized after the ozone addition, forming aged POA, and after 7 h of aging, more than 75% of the original POA was transformed. This process may involve evaporation and homogeneous gas-phase oxidation as well as heterogeneous oxidation of particulate organic matter. The results generally prove that logwood burning emissions are subject of intensive chemical processing in the atmosphere, and the time scale for these transformations is relatively short, i.e., hours."

Comment 2: P4:R28-: The authors chose to investigate cold start (kindling) emissions

using a very dry fuel. Since this a very important study the reader needs to know about the generalizability of the chosen combustion variables. How would the findings transform to fuel addition on glowing embers (as I would suspect is more representative for ambient biomass PM) and fuel with higher moisture content?

Answer 2: Generally, the emission concentrations and characteristics from residential wood combustion (RWC) vary remarkably. Various combustion technologies (boilers, stoves, open fireplaces etc.) and fuels (wood logs, pellets, different species etc.) are used, which all generate different specific particulate and gaseous emissions. In addition, operational practices have large effects on the emissions and fine particle properties. The chosen case in this study reflects typical combustion conditions in the chosen appliance type. However, generalization of these results to all kinds of combustion cases is not possible. Combustion of a single wood batch includes cold start (this occurs in all RWC cases) and also different phases (ignition, flaming combustion, char burning). Our experiments included all three phases and can therefore be regarded as representative although it cannot be generalized for all types of wood-log combustion emissions. The ignition of a new wood log batch during addition of fuel on glowing embers is lacking, as was pointed out by the reviewer. Also the emissions caused by this type of ignition vary widely, depending, e.g., on the temperature of the stove as was shown by by Leskinen et al. 2014. Effects of fuel moisture content on emissions is very complex. Ignition of wet wood is not typically successful and produces high emissions. However, wet fuel can even lower emissions, e.g., in cases of too fast gasification of fuel (e.g. due to hot firebox in masonry heater after the combustion of several batches). In this case, dry wood fuel was chosen because it is generally recommended to use, it is of commercial quality, and it gives more reproducible results when the repetition of tests are performed. We have added some clarifications in the lines 29-31 (page 4) and 1-6 (page 5) to give the reader information on the generalizability of the chosen combustion variables as follows. "In each experiment, 2.5 kg of wood logs (spruce [Picea abies], birch [Betula pubescens] and beech [Fagus sylvatica]) were burned with combustion initiated from 'cold start', i.e., the firebox was at room temperature. Each

combustion experiment included the ignition, flaming combustion and residual char burning phases. Ten logs (∼235 g each) were laid crosswise with 150 g of smaller wood pieces as kindling on top. The combustion was initiated by igniting a batch of wood sticks, serving as kindlings, on top of the wood batch. Low pressure in the stack was adjusted with a flue gas fan and dampers to $12.0 \pm 0.5$ Pa below ambient pressure. The moisture contents of the wood logs were 7.2 %, 7.4 % and 9.0 %, for birch, spruce and beech, respectively (See Reda et al. 2015 for details), which agrees with a kiln dried commercial wood log quality. The ignition of the wood batch was carried out in two different ways: 1) smaller kindling size resulted in faster ignition of the main batch and 2) larger kindling size (double) resulted in slower ignition of the main batch (Table 1). The mass of kindling was 150 g / ignition. The gaseous emissions from the raw exhaust 5 gas were measured online with a multicomponent FTIR analyzer (Gasmet Technologies Inc.), including $CO_2$, CO, NO, $NO_2$, $CH_4$ and 15 typical hydrocarbon compounds (Table S1, Figs. S1 and S2) for wood combustion emissions. In addition, the analyzer was equipped with an $O_2$ sensor.

Comment 3: For example P6R4: The authors make several notes about the relevance for Chinese conditions. I agree biomass combustion emissions are very important for air quality in China. However, for the logic to make sense please comment in the paper if the chosen combustion system (and fuels) are of relevance for Chinese biomass combustion emissions. Answer 3: The reviewer is correct that the studied logwood source is untypical for Chinese conditions. The main point was to explain that there are significant differences in atmospheric chemistry between polluted and non-polluted areas. We modified the line 30 (page 15) and line 1 (page 16) on as follows:

"These are prevalent, e.g., in the commonly observed haze pollution events in Megacities, whose features are very different from those reported for non-polluted aerosol systems in the atmosphere." Comment 4: P6R30: Dual or Tungsten vaporizer? Which was used in the presented results beyond the rBC data? Please state. Please check that the 60 s is enough to get a stable signal without being sensitive to transients

when laser goes on-off. Was there any transients observed or were the results "perfect square waves"?

Answer 4: Thank you for catching this! This is a very important point. The SP-AMS was operated by alternating the laser vaporizer on and off every minute, while always running the heated tungsten vaporizer. To ensure that turning the laser vaporizer on and off did not affect the tuning or the ion signals derived from the tungsten vaporizer (by striking and heating elements in the ion formation chamber), the filament emission was carefully watched and showed little to no variation during operating. The laser light intensity (monitored with a CCD camera) showed that the laser intensity was stable within a few seconds of being turned on. Minor transients were observed in the closed (i.e., background with particle beam blocked), but these do not affect the difference measurements. Given the uncertainties in the laser vaporizer CE values (Willis et al., 2014), the observed differences in the ion signals (specifically increased CO+ and CO2+ signals compared with the tungsten vaporizer) (Canagaratna et al., 2015), and the observation of refractory PM signals (Onasch et al., 2012), we have chosen to use the dual vaporizer mode to quantify and discuss the rBC (and potassium) measurements, whereas all other species (Org, NO3, SO4, NH4, Chl) and associated analyses (i.e., O:C, PMF, etc.) were done using the standard tungsten vaporizer results. To make this clear in the manuscript, we have added the following text: Added to the paragraph at the end of Section 2.2 Measurements of particles and their chemical composition in the chamber "The SP-AMS dual vaporizer mode was used to characterize the rBC (and potassium) mass loadings and chemical signatures; all other particulate chemical species (Org, NO3, SO4, NH4, and Chl), specifically including all analyses of OA and SOA, were done using the standard tungsten vaporizer measurements."

Comment 5: P7R8: As I understand it a reduced CE (0.6) was needed compared to the mIE calibration with Regal Black (RB). Thus the sensitivity of the SP-AMS to biomass rBC was lower than for RB. Note that this is opposite to the results by Willis et al. (2014) who found that the sensitivity increased upon coating RB. I do not argue

against the CE of 0.6 for biomass compared to RB, it is similar to published data for wood stoves (Martinsson et al. 2015). However, it should be motivated based on the TEOM comparison or biomass data in literature and not based on the Willis et al. reference

Answer 5: The reviewer is correct that the Willis et al. (2014) reference is not the source of the CE value. Sorry for the confusion. The SP-AMS collection efficiency (CE) for rBC mass loadings of 0.6 is based on a direct comparison to elemental carbon (EC) concentrations analyzed from filter samples (20 min) during the stabilization phase of the experiments (i.e., before oxidation). The measured CE's for all experiments were consistent (< 5% difference). Elemental carbon concentrations were analyzed from filter samples by a thermal-optical method. TEOM comparison results are presented in Table 2 (PM/Total) where PM is TEOM concentration and Total is sum of AMS masses. To make this clearer in the text, we modified the wording in the final paragraph in section 2.2. "The aerosol mass concentrations were also corrected for the particle collection efficiency (CE). The CE values can be smaller than 1 because of losses in the aerodynamic inlet or on the vaporizer (tungsten) or increasing divergence in the particle beam (laser vaporizer). Collection efficiencies (CEs) of 0.5 and 0.6 for tungsten (Canagaratna et al., 2007) and the laser vaporizer (Willis et al., 2014), respectively, were applied in the concentration calculations. The laser vaporizer CE (0.6) was determined via direct comparison with off-line elemental carbon (EC) concentrations analyzed from filter samples (20 min) during the stabilization phase of the experiments (i.e., before oxidation). The SP-AMS dual vaporizer mode was used to characterize the rBC (and potassium) mass loadings and chemical signatures; all other particulate chemical species (Org, NO3, SO4, NH4, and Chl), specifically including all analyses of OA and SOA, were done using the standard tungsten vaporizer measurements. Therefore, all particulate measurements (other than rBC and K) represent the standard AMS operationally defined nonrefractory portion of these chemical species (Jimenez et al., 2003). These CEs were verified by comparisons with Tapered Element Oscillating Microbalance (TEOM, Thermo Scientific, model 1405) total PM1 mass. Any

uncertainties associated with the CEs may affect the absolute concentrations of the measured species and the relative OM-to-rBC ratios, although they are less likely to affect the SOA formation ratios, which are discussed later." Comment 6: P7R21: So the CE of rBC did not change much upon aging? Please clearly state if this is correct or not. I assume it makes sense since the rBC particles are only thinly coated by aging.

Answer 6: The SP-AMS CE for rBC mass loadings, as determined by comparisons with off-line elemental carbon measurements (thermal optical technique), was observed to increase slightly during aging, due to coating of the rBC with SOA (Willis et al., 2014), and were corrected in the data-analyses. The maximum correction due to the observed CE increase was 15%. With the modified wording to comment 5 above, the answer to comment 6 should be clear in the text (section 2.3 Data analysis). Comment 7: P9R16: It is very good that it is pointed out that the SP-AMS measurement of K is highly uncertain. However, also the quantification of Cl and SO4 requires caution as their chemical state is quite different from ambient observations. Their quantification from SP-AMS data is new for me in biomass combustion emissions. Please add to the methods section how this was done (vaporizer mode, CE, RIE etc). As is stated K salts may be present as external mixtures. If so what would make the rBC free particles vaporize in the W and dual vaporization modes respectively, given their absorption cross sections and vapor pressures? Was there any filter samples to validate the new method to quantify Chl and SO4 in biomass combustion data?

Answer 7: As now, hopefully made clearer in our answer to comment 4, all chemical species other than rBC and K were obtained from the standard vaporizer measurements. Thus, our measurements of the inorganic particulate chemical species (NO3, SO4, NH4, and Chl) relate only to the operationally defined non-refractory portion of these species. In particular, the NO3 shown related to organonitrates (ON) in section 3.4.3. To make this clearer in the text, we have added the following wording into the final paragraph in section 2.2:

"Therefore, all particulate measurements (other than rBC and K) represent the stan-

dard AMS operationally defined nonrefractory portion of these chemical species (Jimenez et al., 2003)."

Comment 8: Figure 2: The small and capital letters a, b, c risks the leader will find it hard to quickly understand the figure. Please change the denotation for the different stages of each experiment to something that is more helpful to the reader.

Answer 8: We agree with the reviewer that letters A, B and C are not ideal for understanding different phases during aging so we have changed to the following: S for stabilization, D for dark aging and U for UV aging in Fig 2. Comment 9: Figure 2b: If the SO4 increase after HONO addition is most likely an artefact then this should be mentioned in the figure caption. However, it is very good that it is mentioned and not sorted away as may sometimes be the case.

Answer 9: The SO4 mass increase after/during HONO addition was an artefact and was caused because a small amount (from 1.5 ppb to 2.5 ppb) of SO2 ended up in the chamber together with the HONO addition. We added text "The SO4 mass increase in Fig 2b was mainly an artefact caused by SO2 which ended up in the chamber together with the HONO addition." in figure 2 caption.

Comment 10: Table 4: double legend, please remove Answer 10: Double legend was removed in Table 4. Comment 11: Table 4: double legend, please remove Comment 11: Table 4: It is very good the age is calculated for OH exposure. Could relevant numbers be derived also for the dark aging?

Answer 11: OH exposures are already calculated in Table 4 for experiments 2B and 3B. D9-butanol concentration didn't decrease during aging in experiment 1B so OH exposure was not possible to calculate. Text "<DL" added for experiment 1B in Table 4 (Dark aging). Comment 12: P10R26: If ozone was added in the "UV-only" experiments, could you then conclude that OH chemistry and not O3 chemistry is responsible for the SOA formation in this case?

Answer 12: In UV-only experiments the SOA formation is caused by both of these effects. To elucidate the role of different aging reactions, PMF was carried out (Figure 5). Based on PMF calculations OH-radical driven reactions (Figure 5: SOA3) are dominating reactions. In fast ignition case (4B) (low VOC to NOX ratio and alkene concentration) effect of ozone chemistry was low.

Comment 13: P11R5: Was the laser vaporizer engaged when calculating the elemental ratios? Please describe in the text. This is critical since the laser may produce a relatively strong CO2+ signal which have been hypothesized to origin from surface oxides on the soot (Corbin et al. 2015). It is not clear to me if this signal should be attributed to the rBC core or the OA coating. Thus, one may speculate that the O:C would be significantly higher for dual vaporisers than W-vaporiser. Please comment in the text the O:C for fresh emissions with dual and W vaporizer, respectively.

Answer 13: As noted in comment 4 above, we agree with the reviewer about the confusion here and hope that our answers to the above comments have address this issue. The elemental ratios were calculated from the standard tungsten vaporizer data (i.e., laser vaporizer was off) to avoid these issues. While comparisons of the two vaporizer modes (dual and tungsten) are of interest and we spent time investigating these, we decided to not focus on this topic in the current paper.

Comment 14: Figure 5: Please change the names of the POA and SOA factors to represent what they are tracing, it would make it much easier to convey the messages to the reader. Add info about fast and slow ignition to the figure, this is currently missing.

Answer 14: The names of the POA factors in Figure 5 are changed to POA1: biomass burning OA (including PAH), POA2: hydrocarbon-like OA and the names of the SOA factors are changed to SOA1: formation by ozonolysis, SOA2. formation by NO3/RO2 (secondary organonitrates), SO3-formation by OH radical to represent their origin as referee suggested. Information about fast and slow ignition is added to figure caption as followed

[Figure]

Comment 15: P14R2: Yes possibly toluene, but Xylenes commonly have a much higher f43/f44 ratio than found here. What about Naphthalene? Another group of important SOA-precursors are phenols. Please comment about their potential contribution to the different factors based on their reactivity towards different oxidants. An important paper on the precursors responsible for SOA formation was recently published by Bruns et al. (2016). Please add this reference and compare your conclusions with their findings.

Answer 15: We thank for the relevant comment. The new article by Bruns et al. (2016) was considered and based on this study some of the potential compounds were named. Phenols / methoxyphenols are known to be very abundant in wood combustion smoke and potential important SOA precursors. We modified the line 1-4 (page 14) and line 1 (page 16) on as follows:

"The potential precursors for this factor are likely aromatic compounds, which have typically slow reaction rates in the presence of ozone. These may include for example toluene, naphthalene, benzene and benzaldehyde which were recently identified as some of the major SOA precursors for a wood stove (Bruns et al., 2016). This conclusion is in agreement with the clearly higher SOA3 mass in the slow ignition case, which contained higher amounts of aromatic compounds in the primary emissions, than in the fast ignition case (Fig. S2)."

Comment 16: Figure 6: It would be important to include the rBC emission factors in this figure (on a separate axis) so it becomes clear for the reader that these aerosols are always rBC dominated.

Answer 16: rBC emissions were added in figure 6 (below) as referee suggested.

Comment 17:P14R9: POA seems to contain very low PAH contribution, why would then PAH degradation by UV be a major course of the reduction of this factor with UV? Answer 17: We fully agree with the question raised by the reviewer. PAH concentrations were low because of efficient burning conditions, so PAH degradation did not dominate the decreasing of POA1 factor when UV light were switched on (Figure S5). It is likely

[Figure]

that a combination of transformation of several compounds as well as partitioning of semi-volatile organic compounds between the vapor and condensed phases caused the observed reduction of this factor. We modified the lines 9-16 (page 14) as follows:

"Both POA1 (BBOA factor including PAH) and POA2 (HOA factor) were higher during the slow ignition than the fast ignition experiments. POA1 decreased mainly after the UV lights were switched on while POA2 was found to decline also via dark aging. The decrease of primary organic matter upon aging can be induced by partitioning of semi-volatile compounds combined with chemical reactions both in vapor and particulate phases. The PAH compounds (Dzepina et al., 2007) included in the POA1 were observed in low concentrations. Total PAH ranged from 0.1 to 0.5 ïA■g m-3 (Fig. S5) which is about 1 % of total OA.

References

Bruns, E. A, El Haddad, I., Slowik, J. G., Kilic, D., Klein, F., Baltensperger, U. and Prevot, A. S. H.: Identification of significant precursor gases of secondary organic aerosols from residential wood combustion, Scientific Reports 6, 27881, doi:10.1038/srep27881, 2016.

[Figure]

[Figure]

Figure 5: Evolution of OA during aging based on PMF calculations: (a) Exp. 1B (fast ignition), (b) Exp. 3B (fast ignition), (c) Exp. 4B (fast ignition), (d) Exp. 2B (slow ignition), and (e) Exp. 5B (slow ignition). The stabilization phase is marked as S, dark aging as D, UV aging as UV, and the HONO addition as H in the upper panel.

**Fig. 1.**

[Figure]

Figure 6: Aged and non-aged POA emission together with SOA at the end of dark (A) and UV aging (B) periods based on PMF calculations (Exp. 1B-5B) as well as (b) refractory black carbon (rBC) emission.

**Fig. 2.**

---

## Author Comment (AC2) · 2 Sep 2016

We thank the referee for a thorough review of our manuscript. The referee comments were very valuable and we believe that addressing the issues raised by the referee will considerably improve the manuscript.

Major comments:

Comment 1, Given that a substantial amount of discussion in the manuscript is made within the context of fast vs slow ignition, the authors should dedicate a paragraph in the results section to discuss the main differences between the two methods in terms

of emission profiles in order to help and guide the reader throughout the discussion.

Answer 1: We extended the discussion about fast vs slow ignitions (lines 16-21, page 9) as referee suggested including an additional figure to Supplements (Fig. S7).

"Furthermore, it was observed in the slow ignition cases that the emissions of non-methane VOCs (NMVOC) show a distinct peak at about 8 – 10 minutes after ignition (Fig. S7), whereas fast ignition produced substantially lower peak at around 5 minutes after ignition. The ignition method had also a strong effect on the combustion rate of the whole batch. In the experiments 2B and 5B there were still yellow flames after 35 minutes (considered as a length of the batch), while in the experiments 1B and 4B the flaming phase ended already at around 28 minutes from ignition and the rest of the combustion process included only char burning without visible flames." The whole chapter 3 (lines 12-16 + new text) without potassium results were moved in the beginning of Chapter 3.1 (Primary emissions).

Comment 2: PMF Factors: The SOA2 factor suggested to be a result of NO3 chemistry should only be important during dark ageing experiment. The fact that the concentration of this factor remain high (or even slightly increase) in Figure 5 suggest that other oxidation sources are important to its formation or that the light source in the chamber does fully represent day time chemistry in the troposphere. This should be discussed and clarified in the revised manuscript. On a related note, the correlation between SOA2 and NO3 mass in Figure S5 should be supplemented by more information showing (perhaps a lack of) relationship between nitrate and SOA1 and SOA3 in order to confirm the argument suggested by the authors.

Answer 2: We thank for the relevant comment. The formation of organic nitrate factor (SOA2) is through two channels: One is the one as the reviewer suggested through the NO3 radical oxidation in case of excessive NOx and O3 in dark:

NO2+O3->NO3 (radical) NO3+VOC->ON The other channel is though photo-chemistry via reactions of peroxy radical (RO2) with NO (Atkinson et al., 2000).

RO2+NO->RONO2 (ON) This most likely explains why we are seeing the formation of SOA2 after UV was switched on. We added new chapter in page 23 right after Chapter 3 with the following text:

"The formation of secondary organic nitrate factor is through two channels: One is through the NO3 radical oxidation in case of excessive NOx and O3 in dark experiments (described in previous chapter) and the another channel is through photochemistry via reactions of peroxy radical (RO2) with NO (Atkinson et al., 2000). This most likely explains why we are seeing the formation of SOA2 after UV was switched on in presence of high NO in experiments 4B (Figs.5d)." New figure S8 was added to clarify the good correlation between nitrate and SOA2 during dark aging in which correlations between SOA1, SOA2 and SOA3 vs. NO3 are now presented (updated Fig. S5i).

Here are the referee minor comments followed by replies:

Comment 1: Page 1, line 21-22: statement about substantial contribution from SOA from small scale wood combustion to global atmospheric PM matter needs to be supported and referenced in the introduction. This issue is more likely to have local/regional impact. In either case, references should be added to support a correct statement.

Answer 1: The reviewer is correct that contribution of small-scale combustion emitted SOA to global atmospheric PM is not well-defined. We modified Abstract (Page1, line 20-22) as follows

"Organic aerosols (OA) derived from small-scale wood combustion emissions are not well represented by current emissions inventories and models, although they contribute substantially to the atmospheric particulate matter (PM) levels. The secondary organic aerosol (SOA) fraction of the organic emissions is formed via. . ."

Also new references were added to add information about the impact of residential

combustion emissions (Butt et al., 2016) (Introduction, Page 4, line 12).

"Small-scale combustion emissions have received insufficient attention until today, although this type of emission is known to be among the largest sources of OM (organic mass) and climate-active BC (e.g., Bond et al.,2004; Denier van der Gon et al., 2015; Butt et al., 2016)."

Comment 2: Page 1, line 24: replace "deployed" with "used", "employed" or "utilised"

Answer 2: "deployed" was replaced with "utilized" in Page 1, line 24.

Comment 3: Page 2, line 21: the phrase "and internally mixed" should be removed from this sentence. The mixing state of these components is not universal and it is known to change depending on source and conditions.

Answer 3: The phrase "internally mixed" was removed (Page 2, lines 20-22).

"The main properties of soot particles, which are mixtures of elemental carbon, organic matter and inorganic species (ash), may change significantly due to coatings formed by atmospheric aging of emissions."

Comment 4: Page 3, line 13: change "generateproducts" to "generate products"

Answer 4: "generateproducts" is changed to "generate products (Page 3, line 13) Comment 5: Page 6, line 3-4: What is the relevance of the extra statement on OH concentrations in China to this study?

Answer 5: Extra statement is not necessary so lines 2-4 (Page 6) are modified as follows

"The hydroxyl (OH) radical is one of the main reactive species in the atmospheric boundary layer; peak daytime OH radical concentrations are in the range of (1-10) ïĆť 106 molec. cm-3 (e.g., Atkinson et al., 2000; Huang et al. 2014)."

Comment 6: Page 7, line 15: References should be made to two very recent and

studies in this area by (Ye et al., 2016 and Krechmer et al., 2016).

Answer 6: New references were added as referee suggested (Page 7, line 15).

"Particle wall losses (WL correction) have been characterized for decades (e.g., Mc-Murry and Rader, 1985), although the estimation of losses of semi-volatile species remains challenging (e.g., Zhang et al., 2014; Kokkola et al., 2014; Krechmer et al., 2016; Ye et al., 2016)."

Comment 7: Page 10, line 21-22: The authors should comment on whether or not the small increase in SO2 from 1.5ppb to 2.5ppb from HONO addition is sufficient to explain the increase in SO4 particle mass from about 1 to 5 $\mu$g/m3 shown in Figure 2b?. A comment about the nature of the sulphate particles should be made given the remark made in the manuscript about a "lack of base such as NH3" (page 13, line 21).

Answer 7: When 1 ppb of SO2 is oxidized about 4 $\mu$g m-3 sulfuric acid is formed, assuming a full conversion, so relatively small changes in gas phase SO2 can explain the observed sulfate increase. The SO4 mass increase after HONO addition was an artefact and was caused because a small amount of SO2 ended up in the chamber together with the HONO addition.

Logwood emitted primary sulfate appears mainly in form of K2SO4 (Torvela et al., 2014) (Page 9, line 21). Secondary sulfate exists most likely as sulphuric acid due to the absence of base such as NH3.

Following modification was added in text (Page 10, lines 21-23)

"The SO4 mass increased because a small amount (from 1.5 ppb to 2.5 ppb) of SO2 ended up in the chamber together with the HONO addition. This secondary sulfate existed probably as sulphuric acid due to the absence of base such as NH3. This was indicated by the very low ammonium ion concentrations."

Comment 8: Page 13, line 16-18: The direct connection made by the authors between the "level of oxidation" and "volatility" of the different types of SOA is not really supported by any volatility measurements in this work. Although it is true that other papers in the literature have made such association for other types of particles (e.g. ambient), it is advised that these statements are either supported directly by measurements or to be appropriately toned down.

Answer 8: We fully agree with the referee that "volatility" of the different types of SOA are not supported by any measurements in this work and we modified text as follows (page 13, lines 15-18) "The oxidation level of SOA1 was clearly lower than SOA2 and SOA3 (Table 6), suggesting that the oxidation products of ozonolysis from logwood combustion are less oxidized than the oxidation products of NO3 or OH radicals, which agrees with earlier findings (e.g., Chhabra et al., 2010).

Comment 11: Page 14, line 9-10: The comment regarding a sharp decrease in POA1 after the UV lights were switched on is not consistent with data shown in Figure 5; none of the plots in this figure show a sharp decrease in POA1. This should be clarified and revised.

We agree that sharp decrease of POA1 is not visually evident based on the Fig.5. However, Fig.S5 shows the decrease of POA1 in the UV-aging phase (Supplement information). Nevertheless, to clarify the POA chapter (age 14, line 9-10) we modified the text regarding POA1 (page 14) as follows:

"Both POA1 (BBOA factor including PAH) and POA2 (HOA factor) were higher during the slow ignition than the fast ignition experiments. POA1 decreased mainly after the UV lights were switched on while POA2 was found to decline also via dark aging. The decrease of primary organic matter upon aging can be induced by partitioning of semi-volatile compounds combined with chemical reactions both in vapor and particulate phases. The PAH compounds (Dzepina et al., 2007) included in the POA1 were observed in low concentrations. Total PAH ranged from 0.1 to 0.5 $\mu$g m-3 (Fig. S5a) which is about 1 % of total OA.

References

Bond, T. C., Streets, D. G., Yarber, K. F., Nelson, S. M., Woo, J. H., and Klimont, Z.: A technology-based global inventory of black and organic carbon emissions from combustion, J. Geo-phys. Res.-Atmos., 109, D14203, doi:10.1029/2003jd0036970,2004.

Butt, E. W., Rap, A., Schmidt, A., Scott, C. E., Pringle, K. J., Reddington, C. L., Richards, N. A. D., Woodhouse, M. T., Ramirez-Villegas, J., Yang, H., Vakkari, V., Stone, E. A., Rupakheti, M., Praveen, P. S., van Zyl, P. G., Beukes, J. P., Josipovic, M., Mitchell, E. J. S., Sallu, S. M., Forster, P. M., and Spracklen, D. V.: The impact of residential combustion emissions on atmospheric aerosol, human health, and climate, Atmos. Chem. Phys., 16, 873–905, doi:10.5194/acp-16-873-2016, 2016.

Krechmer, J. E., Pagonis, D., Ziemann, P. J., and Jimenez, J. L.: Quantifi- cation of Gas-Wall Partitioning in Teflon Environmental Chambers Using Rapid Bursts of Low-Volatility Oxidized Species Generated in Situ, Environ. Sci. Technol., 50, 5757–5765, 10.1021/acs.est.6b00606, 2016.

Ye, P., Ding, X., Hakala, J., Hofbauer, V., Robinson, E. S., and Donahue, N. M.: Vapor wall loss of semi-volatile organic compounds in a Teflon chamber, Aerosol Science and Technology, 50:8, 822-834, doi:10.1080/02786826.2016.1195905, 2016.
* * *
[Figure]

[Figure]

Figure S7. Time series from emissions of non-methane VOC and nitrogen oxides and their ratio measured with FTIR from raw flue gas. Dash line indicates a point at fast ignition experiments when visible flames extinguished and only char burning continued.

**Fig. 1.**

[Figure]

Figure S8. PMF factors (a) SOA1, (b) SOA2 and (c) SOA3 vs. particulate nitrate during dark aging and corresponding correlation coefficients ($R^2$) for SOA2.

**Fig. 2.**

---

## Author Response (AR1)

[revised manuscript text omitted]
   | <dl< td=""><td><dl< td=""><td><dl[pt22]< td=""><td>4.6E+6</td><td>5.0E+10</td><td>0.58</td></dl[pt22]<></td></dl<></td></dl<> | <dl< td=""><td><dl[pt22]< td=""><td>4.6E+6</td><td>5.0E+10</td><td>0.58</td></dl[pt22]<></td></dl<> | <dl[pt22]< td=""><td>4.6E+6</td><td>5.0E+10</td><td>0.58</td></dl[pt22]<> | 4.6E+6   | 5.0E+10 | 0.58     |
| 2B   | 6.6E+5                                                                                                                        | 9.2E+9                                                                                              | 0.11                                                                      | 4.9E+6   | 5.2E+10 | 0.59     |
| 3B   | 4.7E+5                                                                                                                        | 7.7E+9                                                                                              | 0.10                                                                      | 2.4E+6   | 2.6E+10 | 0.30     |
| 4B   | -                                                                                                                             | -                                                                                                   | -                                                                         | 3.4E+6   | 5.1E+10 | 0.58     |
| 5B   | -                                                                                                                             | -                                                                                                   | -                                                                         | 4.6E+6   | 6.6E+10 | 0.77     |

|      | -    | -    |       |       |        |         |       |      |       |      |       |      |
|------|------|------|-------|-------|--------|---------|-------|------|-------|------|-------|------|
| Exp. |      |      |       |       | Dark/U | V aging |       |      | Total |      |       |      |
|      | O:C  | H:C  | OM:OO | C OS  | O:C    | H:C     | OM:OC | OS   | O:C   | H:C  | OM:OC | OS   |
| 1A   | 0.41 | 1.55 | 1.69  | -0.73 | 0.83   | 1.31    | 2.21  | 0.35 | -     | -    | -     | -    |
| 2A   | 0.35 | 1.50 | 1.60  | -0.80 | 0.75   | 1.29    | 2.11  | 0.21 | -     | -    | -     | -    |
| 3A   | 0.53 | 1.48 | 1.84  | -0.42 | 0.77   | 1.43    | 2.15  | 0.11 | -     | -    | -     | -    |
| 1B   | 0.55 | 1.52 | 1.86  | -0.42 | 0.81   | 1.33    | 2.19  | 0.29 | 0.83  | 1.31 | 2.21  | 0.34 |
| 2B   | 0.53 | 1.50 | 1.84  | -0.44 | 0.74   | 1.35    | 2.10  | 0.13 | 0.82  | 1.30 | 2.20  | 0.33 |
| 3B   | 0.47 | 1.56 | 1.75  | -0.62 | 0.78   | 1.33    | 2.15  | 0.22 | 0.96  | 1.29 | 2.39  | 0.63 |
| 4B   | 0.55 | 1.51 | 1.87  | -0.41 | 0.73   | 1.38    | 2.09  | 0.08 | -     | -    | -     | -    |
| 5B   | 0.51 | 1.52 | 1.81  | -0.50 | 0.75   | 1.36    | 2.11  | 0.14 | -     | -    | -     | -    |

Table 5: Overview of the OA elemental analysis at the end of each experimental stage (stabilization, dark aging, and UV aging); dark aging results are shown in bold.

| Factor | O:C  | H:C  | OM:OC | OS    |
|--------|------|------|-------|-------|
| POA1   | 0.46 | 1.58 | 1.74  | -0.67 |
| POA2   | 0.45 | 1.58 | 1.74  | -0.69 |
| SOA1   | 0.60 | 1.40 | 1.92  | -0.20 |
| SOA2   | 0.86 | 1.26 | 2.26  | 0.47  |
| SOA3   | 0.95 | 1.28 | 2.38  | 0.62  |
| -      |      |      |       |       |

Table 6: Elemental composition of the PMF factors POA1, POA2, SOA1, SOA2 and SOA3.

| Exp. | Stabilizatio       | on       | Dark aging         |          | Total |
|------|--------------------|----------|--------------------|----------|-------|
|      | NO/NO 2 | Emission | NO/NO 2 | Emission |       |
| 1A   | 5.6                | 31.2     | 5.9                | 24.6     | 55.8  |
| 2A   | 5.3                | 27.4     | 8.3                | 47.7     | 75.1  |
| 3A   | 10.0               | 22.5     | 10.0               | 28.5     | 51.0  |
| 1B   | 6.9                | 34.9     | 7.9                | 21.5     | 56.4  |
| 2B   | 5.1                | 26.7     | 5.8                | 24.3     | 51.0  |
| 3B   | 5.4                | 22.7     | 6.5                | 25.7     | 48.4  |
| 4B   | 5.5                | 29.3     | -                  | -        | 29.3  |
| 5B   | 4.5                | 19.8     | -                  | -        | 19.8  |

Table 7: Organonitrate (ON) emissions (mg kg-1 fuel) from logwood combustion at the end of experimental periods (stabilization and dark aging) calculated using Eq. (2) and (3) and corresponding NO-to-NO2 ratios.

Figure 1: Experimental setup in the ILMARI combustion research facility at the University of Eastern Finland.